# Experimental Study and Finite Element Calculation of the Behavior of Special T-Shaped Composite Columns with Concrete-Filled Square Steel Tubulars under Eccentric Loads

**Quan Li** [1]**, Zhe Liu** [1,2,*] **, Xuejun Zhou** [1] **and Zhen Wang** [3]

1 School of Civil Engineering, Shandong Jianzhu University, Jinan 250101, China
2 Shandong Winbond Construction Group Co., Ltd., Weifang 262500, China
3 School of Transportation & Civil Engineering, Shandong Jiaotong University, Jinan 250357, China
* Correspondence: liuzhe0624@126.com

**Abstract:** Special T-shaped composite columns with concrete-filled square steel tubulars have good restraint on internal concrete, are convenient to process, have a high bearing capacity and good mechanical properties, and can increase the aesthetics of the building and the utilization rate of indoor space. Theoretical analysis, experimental study, and numerical simulation of the eccentric compression performance of the special-shaped column are carried out. Taking the specimen length, eccentric distance, and eccentric direction as test parameters, nine specimens with different slenderness ratios were designed to carry out eccentric compression tests. The eccentric compression performance was numerically simulated and analyzed by the general finite element software ABAQUS. The results show that the short column mainly suffers section strength failure, while the middle and long columns mainly suffer bending instability failure without torsional deformation. The degree of influence of the test parameters decreases in turn according to the eccentric distance, eccentric direction, and length of the specimen; there is no weld cracking phenomenon, and the square steel pipes can work together. The finite element calculation results are in good agreement with the experimental and theoretical values.

**Keywords:** concrete-filled square steel tube; special T-shaped composite column; eccentric compression; mechanical properties; finite element calculation; experimental study

## 1. Introduction

There are many forms of special-shaped sections of CFSTs, mainly including ordinary special-shaped sections (Figure 1), special-shaped sections with restrained tie rods (Figure 2), special-shaped sections with built-in stiffeners (Figure 3), multi-chamber special-shaped sections (Figure 4), combined special-shaped sections (Figure 5), and lattice special-shaped sections (Figure 6). At present, the research on CFST special-shaped columns mainly focuses on their mechanical properties.

For ordinary special-shaped concrete-filled steel tubular columns, Shen et al. [1] and Lei et al. [2] studied the eccentric compression performance, hysteretic performance, and joint seismic performance of L-shaped and T-shaped CFST columns. The plastic deformation and energy dissipation capacity of the nodes are better, and a simplified formula for the bearing capacity of L-shaped and T-shaped CFST columns is proposed. Li et al. [3] and Zuo et al. [4] designed special-shaped columns with restrained bars, and studied the axial and bias performance of the components through axial compression and bias tests. The restrained bar changes the buckling mode of the steel tube, which can increase the bearing capacity of the CFST column and improve the ductility. Additionally, the calculation formula of the axial bearing capacity is established. Wang et al. [5] studied the hysteretic performance of T-shaped CFST columns with built-in stiffeners. Research shows that the width-to-thickness ratio mainly affects the sequence of peak loads and the

decreasing trend of bearing capacity. Stiffeners can limit the deformation of weak parts, delay local buckling, and make the steel pipe and concrete work together. Tu et al. [6] conducted experimental research on the axial compression performance of multi-chamber CFST T-shaped CFST columns and found that the failure of medium and long columns is due to overall bending failure, and multi-chamber CFST columns can enhance the restraint of the section on the concrete. Du et al. [7], Cao et al. [8] and WANG et al. [9] conducted an experimental study on the axial compression and eccentric compression of T-shaped and L-shaped CFSTs. Research shows that the flexural performance and plastic deformation ability of the T-column with CFSTs are better, and the specimens mainly show three types of failure modes: shear failure, local bulging (or cracking), and bending instability. The axial compressive bearing capacity of the specimen has an obvious influence. The strength damage is mainly hoop specimens and welded short columns. The main damage mode of the welded slender column is bending instability damage. Rong et al. [10] and Zhou Ting et al. [11] designed lattice-type T-shaped, cross-shaped, L-shaped CFST composites with special-shaped columns to study their mechanical properties. The results show that the final failure form of the compression-bending specimen is the overall bending instability failure, the steel strength has an obvious effect on the compression-bending bearing capacity of the special-shaped column, and the affixed plate has a better restraint effect on the single-limb column.

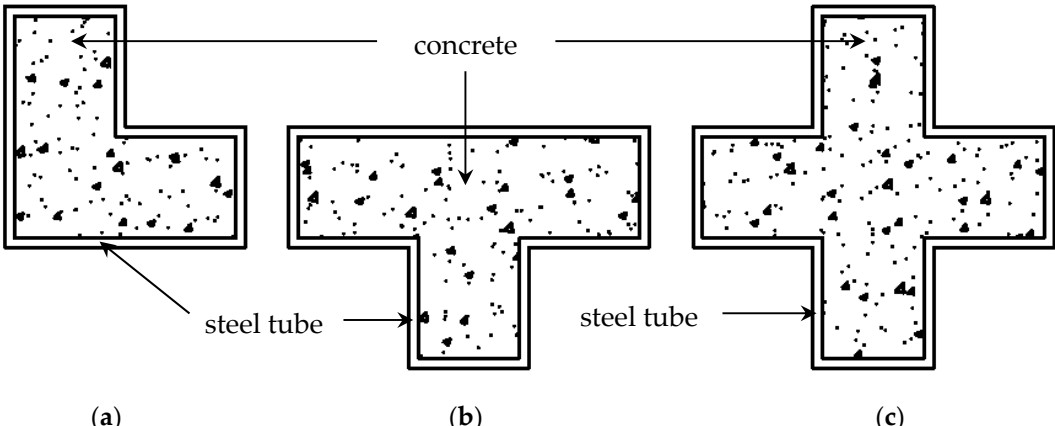

**Figure 1.** Ordinary special-shaped section: (**a**) L-shaped; (**b**) T-shaped; (**c**) cross.

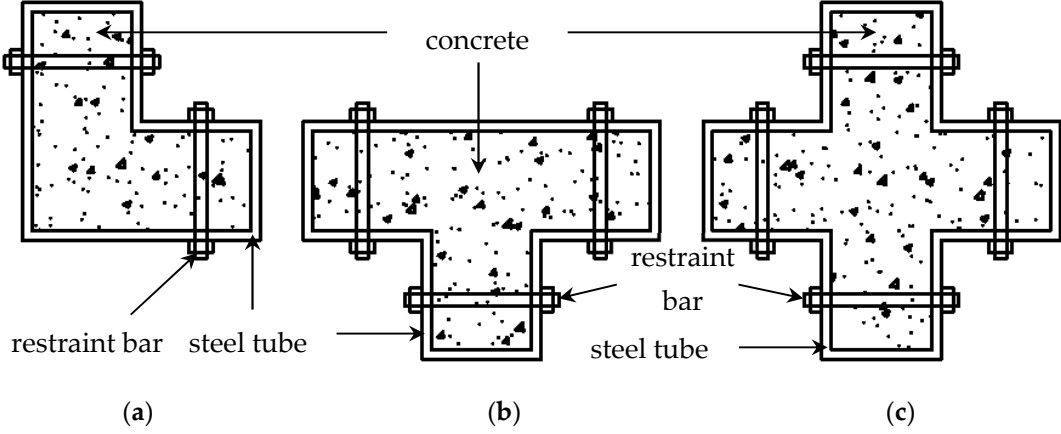

**Figure 2.** Profiled sections with restrained bars: (**a**) L-shaped; (**b**) T-shaped; (**c**) cross.

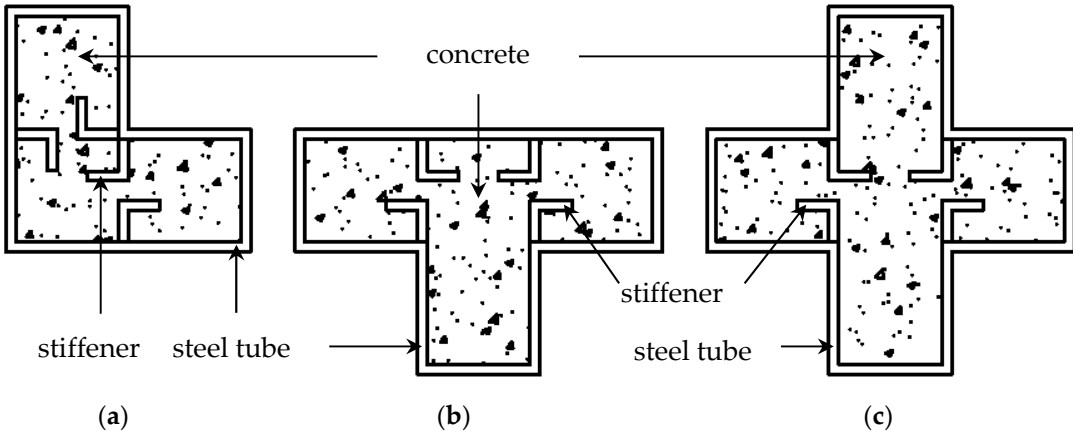

**Figure 3.** Profiled section with built-in stiffeners: (**a**) L-shaped; (**b**) T-shaped; (**c**) cross.

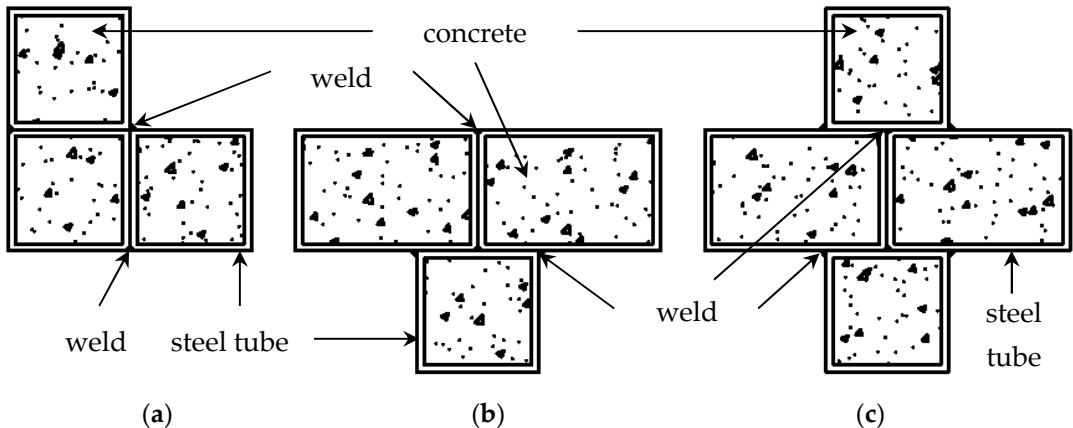

**Figure 4.** Multi-chamber profiled section: (**a**) L-shaped; (**b**) T-shaped; (**c**) cross.

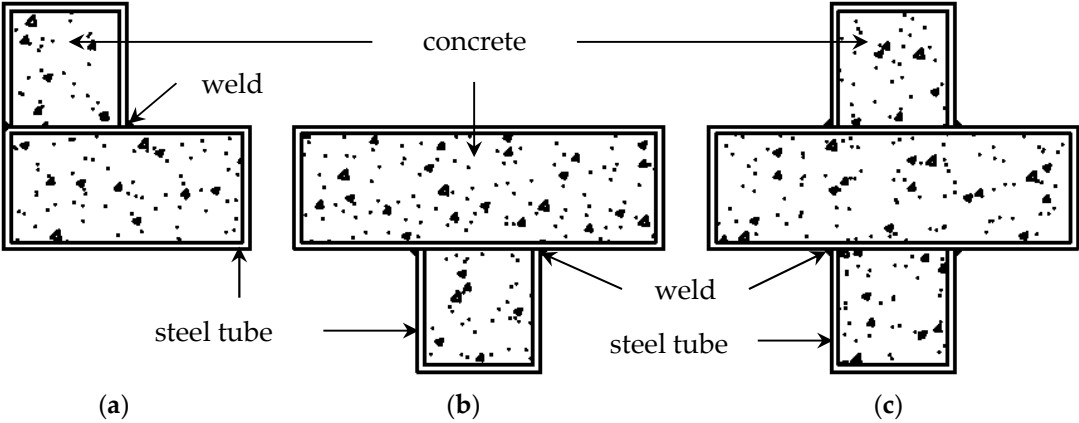

**Figure 5.** Combined special-shaped section: (**a**) L-shaped; (**b**) T-shaped; (**c**) cross.

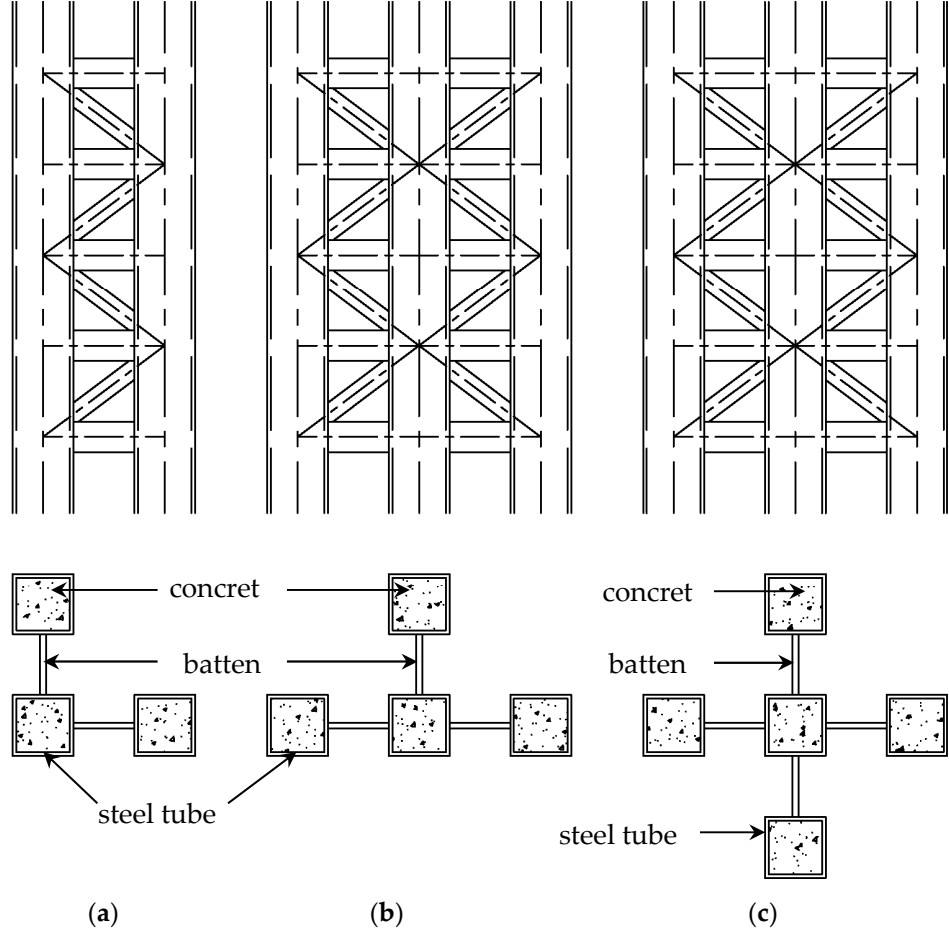

**Figure 6.** Lattice-shaped section: (**a**) L-shaped; (**b**) T-shaped; (**c**) cross.

Zeghiche, J. et al. [12], Ghannam S. [13], Ahiwale D. et al. [14], Achuthan, P. et al. [15], Umamaheswari, N. et al. [16] and Gupta, P. K. et al. [17] conducted axial and eccentric compression tests on CFST columns. It was found that the CFST columns were all damaged due to global buckling. The axial bearing capacity of CFST specimens with artificial sand is significantly improved. The axial and biased column bearing capacity test results of unidirectional bending are in good agreement with the calculation results of Eurocode 4. Phan h D. [18], Saleh, S. M. et al. [19], Jayalekshmi, S. et al. [20], Heman, A. M. et al. [21] and Bhartiya, R. [22] et al. conducted a finite element analysis on the axial compression performance of CFST columns. Studies have shown that increasing the concrete grade leads to a decrease in the ductility of the composite column. Steel yield strength and tube wall thickness contribute to ultimate strength. Restrained tie rods significantly increase the strength and ductility of the CFST column. Boukhalkhal S H et al. [23] found that CFST columns performed better in terms of ductility, plastic hinge distribution, and appearance order. Lazkani, A. [24], Tam, V. W. et al. [25], Ahmad, S. et al. [26], Mujdeci, A. et al. [27], Esmaeili Niari, S. et al. [28], De Azevedo et al. [29], Malathy, R. et al. [30] and Portolés, J. M. et al. [31] replaced the concrete filled in the steel tube with a different material. Studies have shown that expansion agents can improve the strength, ultimate strain, and ductility of recycled aggregate CFST columns. The bearing capacity of the concrete-filled rubber-filled steel tubular column section decreases with the increase in rubber content, but its ductility increases significantly. Adding 20% iron filings via sand weight increases the initial stiffness. The bearing capacity of recycled aggregate concrete-filled steel tubular columns is higher than that of ordinary columns. The impact of high-strength or ultra-high-strength concrete on axial compressive bearing capacity is more significant than that of eccentric compressive bearing capacity.

The inner corners of ordinary CFST columns with special-shaped sections are weak and cannot effectively constrain the corner concrete. There are many openings in the restraint tie rod which weaken the section of the steel pipe, cause a lot of stress concentration, and make the surface uneven. The built-in stiffening rib plate is inconvenient to process, the welding area is large, and the welding residual stress is large. Multi-chamber special-shaped steel pipes have different cross-sectional dimensions which are difficult to standardize and inflexible in layout. The long side of the combined special-shaped rectangular steel pipe is weak against the concrete. The special-shaped section of the lattice type is cumbersome in form, and the stress situation is complex, which is not convenient for analysis and calculation.

However, there is still a lack of relevant research on the mechanical properties of special-shaped CFST composite columns. After improving and optimizing the special-shaped section of CFSTs, a new type of special-shaped square CFST composite column is proposed. Figure 7 shows special-shaped columns with three cross-sections: L-shaped, T-shaped, and cross-shaped. The square steel tube concrete composite special-shaped column is directly processed and manufactured by the finished square steel tube. The welding forming is simple and convenient, the processing speed is fast, and it is easy to be produced in a factory. It can reduce the difficulty of on-site construction and ensure the quality of the welding seam, avoid the appearance of column edges and corners in the room, standardize the cross-section form, and make the layout flexible. The problem of deformation of the inner corner of the section has been improved. The surface is flat without protrusions and holes, reducing the weakening of the section by the opening. By reducing the aspect ratio of the section, the local stability of the steel tube and the restraining effect of the internal concrete are enhanced.

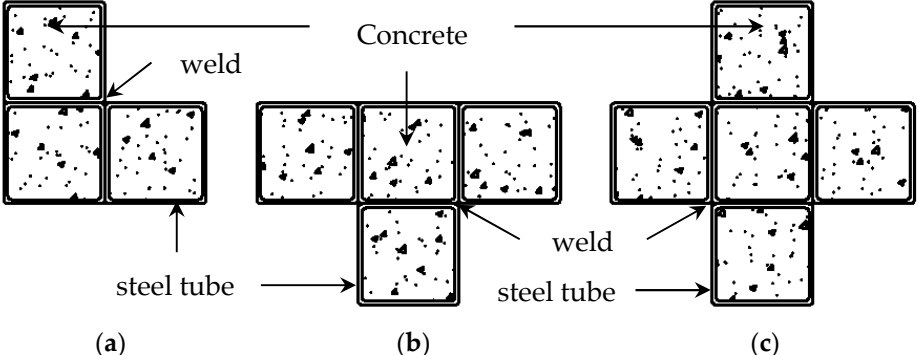

**Figure 7.** Cross-section of CFST composite special-shaped column: (**a**) L-shaped; (**b**) T-shaped; (**c**) cross.

## 2. Experimental Program

### 2.1. Specimen Design and Material Properties

In the T-shaped square steel tube concrete composite special-shaped column, the section size of the hollow square steel tube is 100 mm × 100 mm × 4 mm, as shown in Figure 8. The specimen is composed of CFST special-shaped column members and steel cover plates, and the T-shaped section is formed by four square steel pipes through four fillet welds, as shown in Figure 9. The design parameters of the nine specimens are shown in Table 1.

Table 2 shows the parameters of the steel samples, and the test results of the mechanical properties of the steel are shown in Table 3. The method of converting elastic modulus $E_c$ and concrete axial compressive strength $f_c$ adopts the calculation formula given in the modified stress–strain relationship of the concrete-filled steel tubular with different strength grades under uniaxial compression proposed by Ding et al. [32], where $f_c = 0.4 f_{cu}^{7/6}$, and $E_c = 9500 f_{cu}^{1/3}$. The test results of the mechanical properties of the concrete are shown in Table 4.

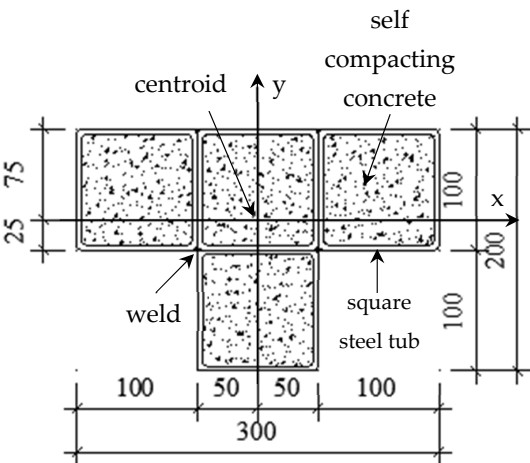

**Figure 8.** Section of T-shaped concrete-filled square steel tubular composite special-shaped column(mm).

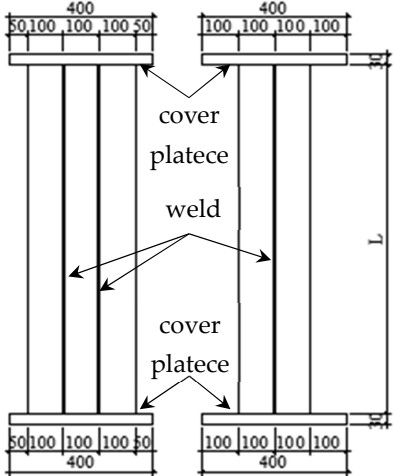

**Figure 9.** Design drawing of the test piece.

**Table 1.** Parameters of the test specimens.

| Specimen | *t*/mm | *L*/mm | *e*/mm | Eccentric Direction | λ |
|----------|--------|--------|--------|---------------------|---|
| T-1 | 4 | 600 | 20 | x+ | 7 |
| T-2 | 4 | 600 | 40 | y+ | 10 |
| T-3 | 4 | 600 | 60 | y− | 10 |
| T-4 | 4 | 1500 | 20 | y+ | 25 |
| T-5 | 4 | 1500 | 40 | y− | 25 |
| T-6 | 4 | 1500 | 60 | x+ | 17 |
| T-7 | 4 | 1800 | 20 | y− | 29 |
| T-8 | 4 | 1800 | 40 | x+ | 20 |
| T-9 | 4 | 1800 | 60 | y+ | 29 |

Notes: *t* denotes wall thickness of steel tube; *L* denotes the length of test piece; *e* denotes eccentricity; λ denotes slenderness ratio. x+ means that the eccentric load action point is in the positive direction of the x-axis; y+ means that the eccentric load action point is in the positive direction of the y-axis; y− means that the eccentric load application point is in the negative direction of the y-axis. $\lambda_x$ and $\lambda_y$ are the slenderness ratios of the components around the x-axis and y-axis, respectively, $l = L/i$, where $i$ is the radius of gyration of the section and $L$ is the length of the test piece, $i = (I_s + I_c E_c / E_s)^{1/2} / (A_s + A_c f_c / f_s)^{1/2}$. The steel design strength grade of the test piece is Q235B, and the concrete design strength grade is C30. $f_s$ is the measured value of the yield strength of the steel; $f_c$ is the measured value of the axial compressive strength of the concrete; $I_s$ and $I_c$ are the section moment of inertia of steel and concrete, respectively; $A_s$ and $A_c$ are the cross-sectional areas of steel and concrete, respectively; $E_s$ and $E_c$ are the elastic moduli of steel and concrete, respectively.

**Table 2.** Parameters of material test specimens.

| Specimen | Number | $a_0$/mm | $b_0$/mm | $L_0$/mm | $L_c$/mm | $r$/mm | Collet Width | $L_t$/mm |
|---|---|---|---|---|---|---|---|---|
| 1–3 | 3 | 4 | 20 | 90 | 115 | 20 | 30 | 350 |

Notes: $a_0$ denotes thickness; $b_0$ denotes width; $L_0$ denotes original gauge length; $L_c$ denotes parallel length; $r$ denotes transition radius; $L_t$ denotes total length.

**Table 3.** Results of steel material properties test.

| Specimen | Thickness/mm | Width/mm | Yield Strength/MPa | Ultimate Strength/MPa | Elongation/% | Yield Strength Ratio |
|---|---|---|---|---|---|---|
| S1 | 4 | 20 | 335.38 | 423.77 | 18.26 | 0.791 |
| S2 | 4 | 20 | 349.63 | 421.13 | 20.87 | 0.830 |
| S3 | 4 | 20 | 348.00 | 427.75 | 19.13 | 0.814 |

**Table 4.** Results of concrete material properties test.

| Group | Number of Test Blocks | Size of Test Blocks/mm | $f_{cd}$/MPa | $f_{cu}$/MPa | $f_c$/MPa | $E_c$/MPa |
|---|---|---|---|---|---|---|
| 1 | 6 | $150 \times 150 \times 150$ | 30 | 44.23 | 33.27 | 33,596.65 |
| 2 | 6 | $150 \times 150 \times 150$ | 30 | 45.33 | 34.24 | 33,872.88 |

Notes: $f_{cd}$ denotes design value of concrete compressive strength; $f_{cu}$ denotes measured compressive strength of concrete cubes; $f_c$ denotes conversion value of concrete axial compressive strength; $E_c$ denotes conversion value of the elastic modulus of concrete.

### 2.2. Test Loading and Measurement

The column hinge is fixed on the end plate, and the boundary conditions at both ends of the specimen simulate the hinge. Figure 10 shows a schematic diagram of loading, and a 500 t pressure testing machine was used for bias testing.

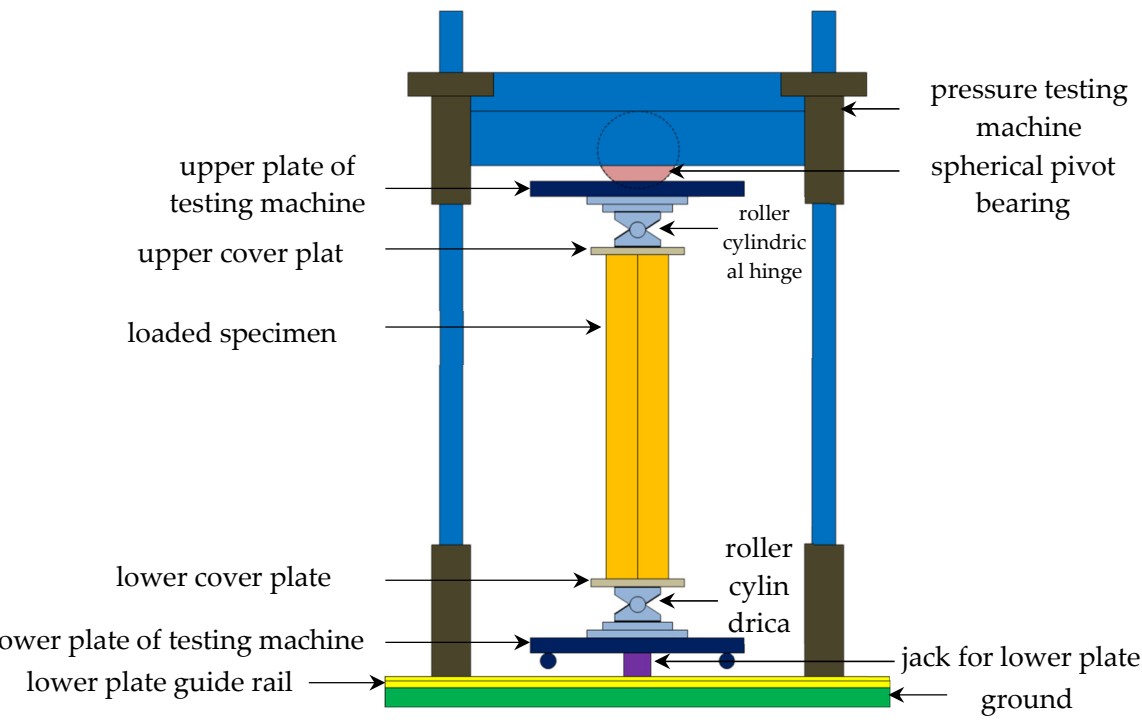

**Figure 10.** Test loading schematic.

The loading regime is shown in Figure 11. In a monotonic static load test, the loading is controlled by displacement. Take 1/10 of the calculated limit displacement as the loading

displacement of each stage. When the load drops to 75% of the ultimate load, stop loading. Unloading adopts force control, and the unloading value of each stage is 1/5~1/20 of the ultimate load. Before the formal test, the specimen shall be preloaded according to 10% of the calculated ultimate load, and the formal loading shall be carried out after checking the state of the specimen and each measuring instrument.

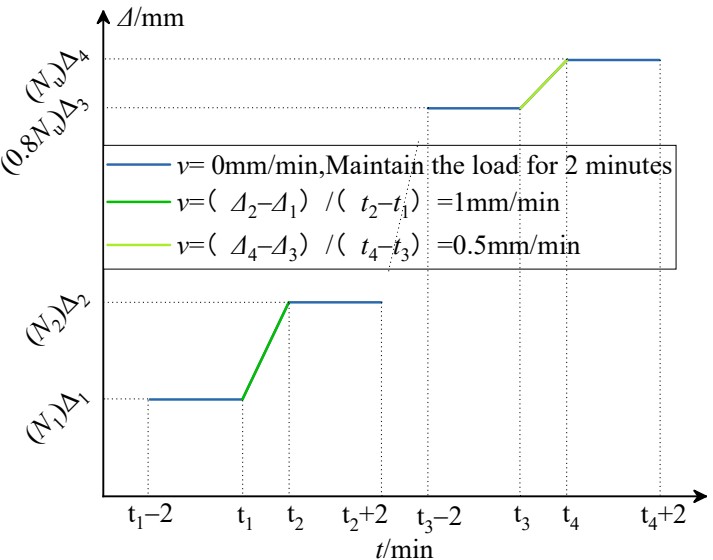

**Figure 11.** Test loading schematic.

Measure the vertical displacement value, the horizontal deflection value of the test piece, and the transverse and longitudinal strain of the steel pipe. Two displacement gauges are set at the lower and upper parts of the specimen to measure the axial displacement of the specimen. Along the height direction of the column, three displacement gauges are set at the quarter points on one side of the test piece to measure the horizontal deflection value of the test piece. Two displacement gauges were installed at the corners of the steel pipe along half of the height direction of the specimen to measure the torsional displacement of the specimen. At 1/2 of the height of the column, a strain (45° in three axes) is arranged on the exposed surface of each steel pipe to measure the longitudinal and transverse strain of the steel pipe. The location of the measuring points is shown in Figure 12.

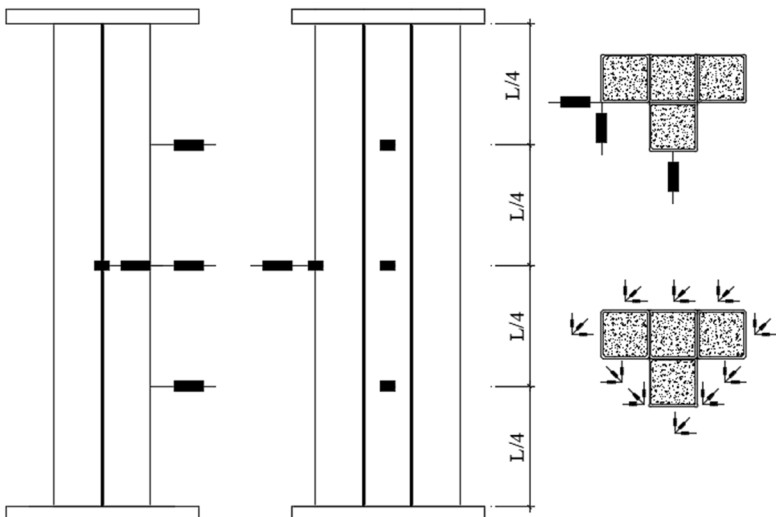

**Figure 12.** Test loading schematic.

## 3. Test Results and Discussion

### 3.1. General Observations and Failure Mode

The failure process of the three short-column specimens is roughly similar; the short-column specimen showed the failure mode of first bulging and then bending, showing the characteristics of strength failure. The short-column specimen did not undergo torsional deformation during the whole loading process. The weld in the short-column specimen did not show visible damage. The welds between the four square steel pipes did not crack, nor did the welds between the square steel pipes and the end plates, and all the welds showed no visible damage. This shows that the four square steel pipes can work together well and bear the force together. The failure process of some short column specimens is shown in Figures 13–15. The failure process of the six long-column specimens is generally similar; the long-column specimen presents a failure mode of bending first and then bulging, showing the characteristics of bending instability failure. There is no torsion phenomenon in the long column specimen during the whole loading process. The welds in the long column specimen also did not show visible damage, the welds between the four square steel pipes did not crack, the welds between the square steel pipe and the end plate did not crack, and all welds did not show visible damage. This shows that the four square steel pipes have good cooperative working performance. Figures 16–21 show the failure process of some long column specimens. The failure results of test pieces 1 to 9 are shown in Figure 22.

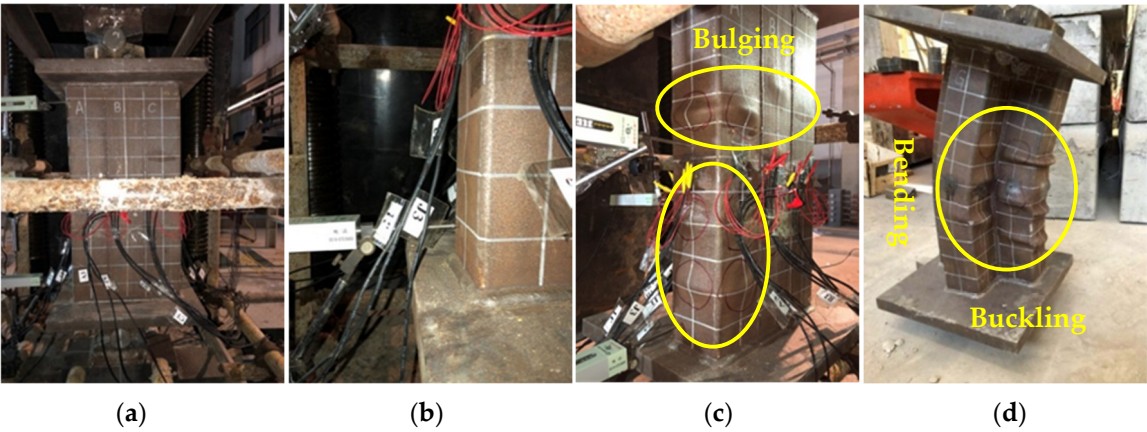

|  (**a**)  |  (**b**)  |  (**c**)  |  (**d**)  |

**Figure 13.** Failure process of specimen T-1: (**a**) Initial state; (**b**) Load status 1; (**c**) Load status 2; (**d**) Damage State.

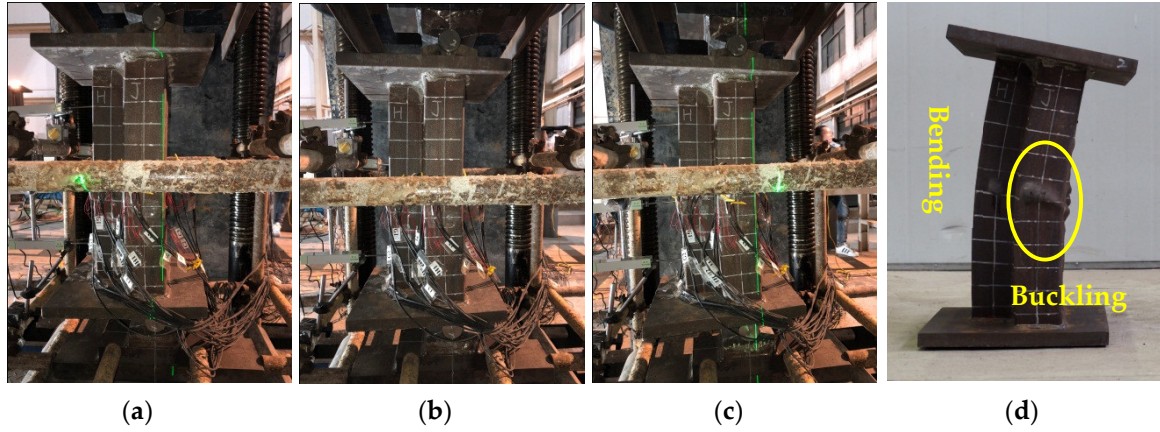

|  (**a**)  |  (**b**)  |  (**c**)  |  (**d**)  |

**Figure 14.** Failure process of specimen T-2: (**a**) Initial state; (**b**) Load status 1; (**c**) Load status 2; (**d**) Damage State.

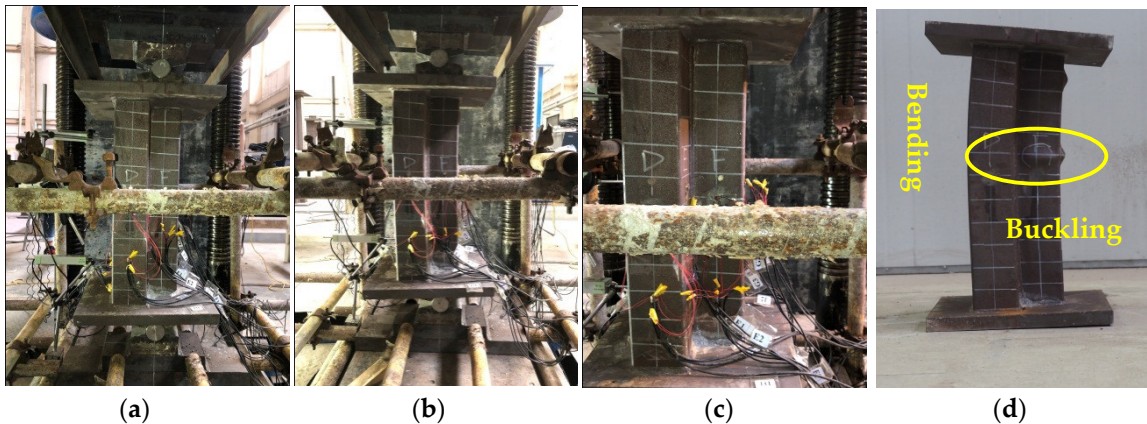

**Figure 15.** Failure process of specimen T-3: (**a**) Initial state; (**b**) Load status 1; (**c**) Load status 2; (**d**) Damage State.

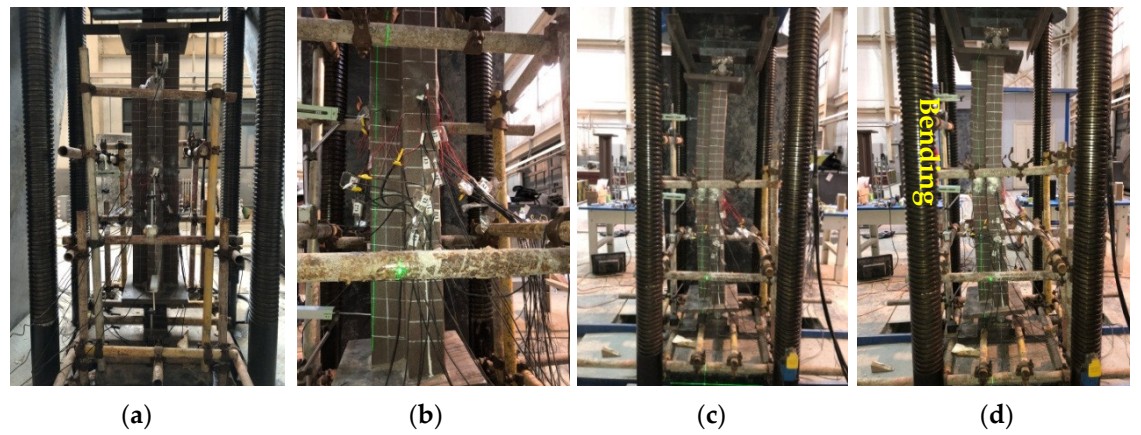

**Figure 16.** Failure process of specimen T-4: (**a**) Initial state; (**b**) Load status 1; (**c**) Load status 2; (**d**) Damage State.

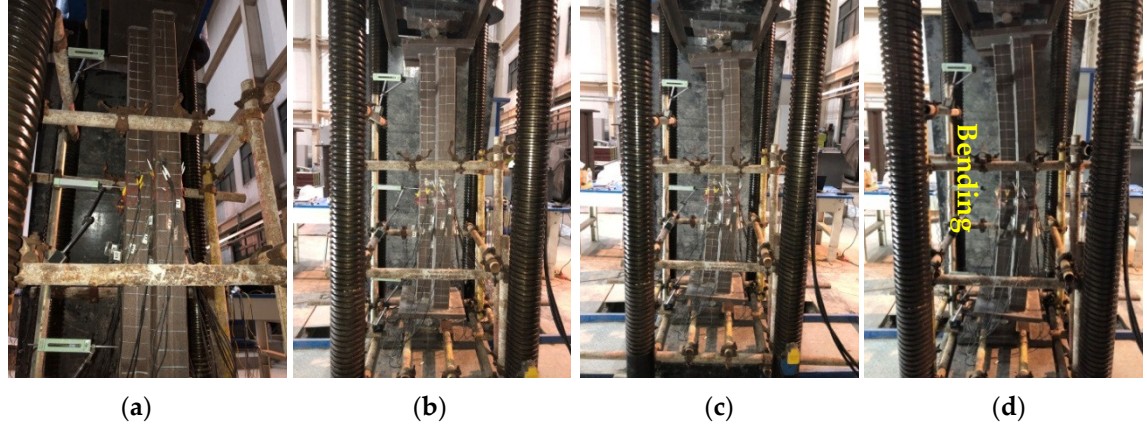

**Figure 17.** Failure process of specimen T-5: (**a**) Initial state; (**b**) Load status 1; (**c**) Load status 2; (**d**) Damage State.

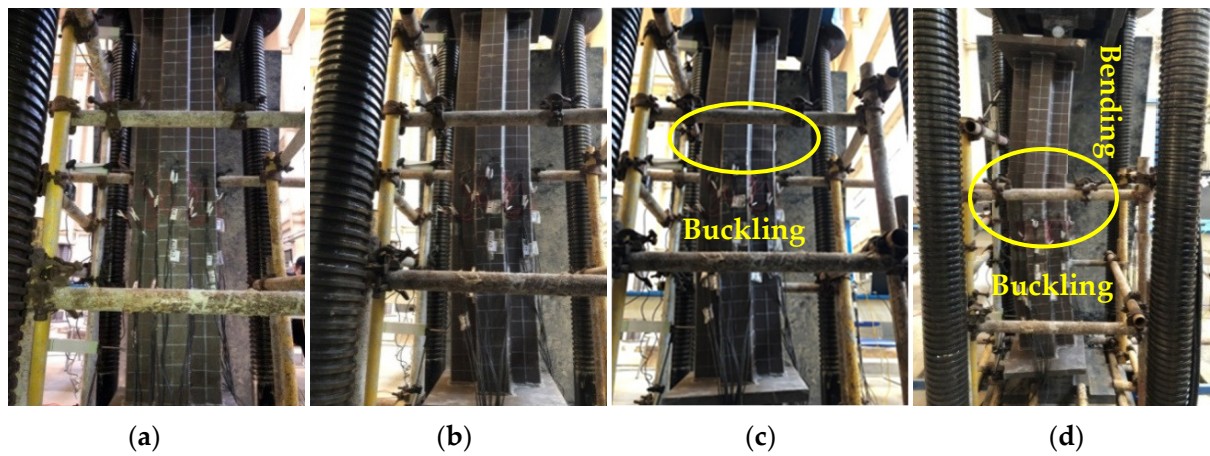

**Figure 18.** Failure process of specimen T-6: (**a**) Initial state; (**b**) Load status 1; (**c**) Load status 2; (**d**) Damage State.

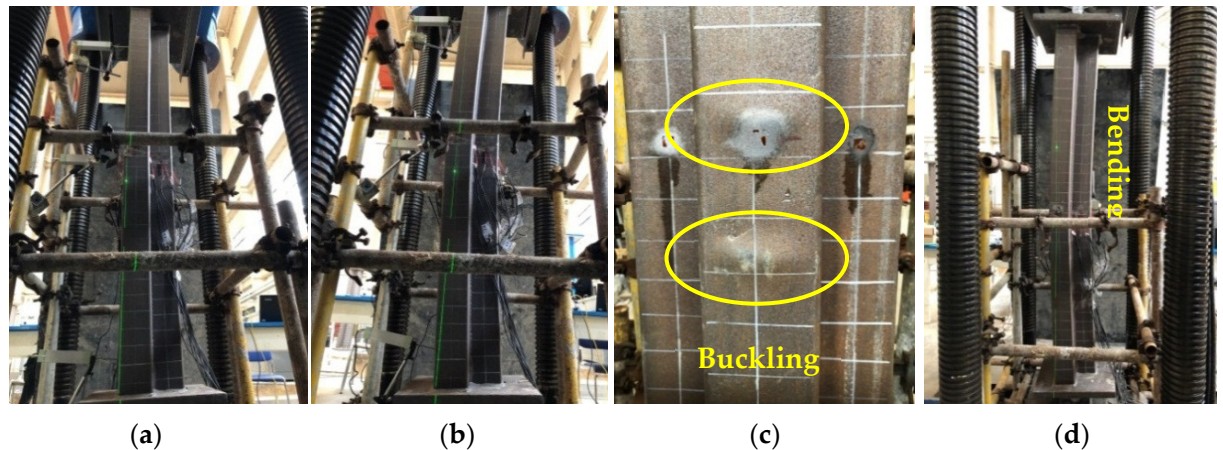

**Figure 19.** Failure process of specimen T-7: (**a**) Initial state; (**b**) Load status 1; (**c**) Load status 2; (**d**) Damage State.

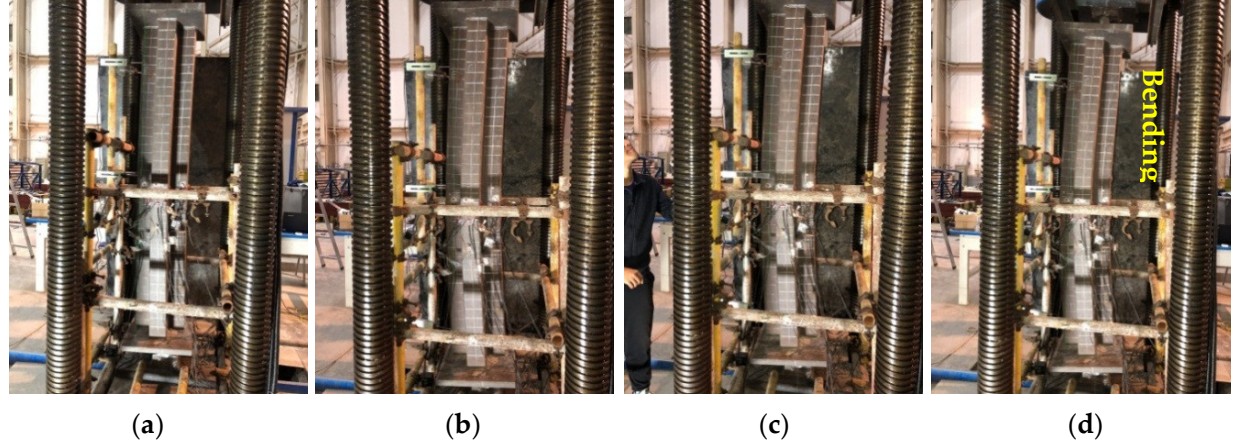

**Figure 20.** Failure process of specimen T-8: (**a**) Initial state; (**b**) Load status 1; (**c**) Load status 2; (**d**) Damage State.

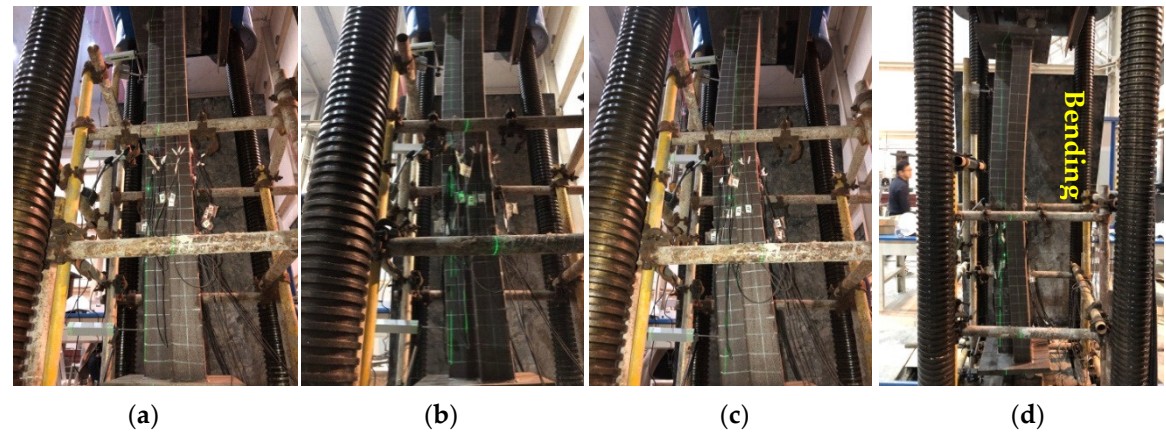

(**a**)            (**b**)            (**c**)            (**d**)

**Figure 21.** Failure process of specimen T-9: (**a**) Initial state; (**b**) Load status 1; (**c**) Load status 2; (**d**) Damage State.

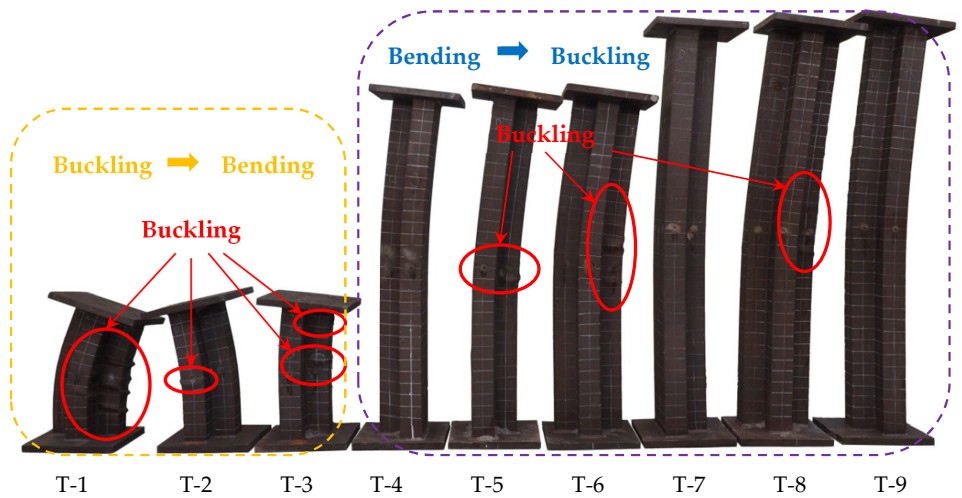

**Figure 22.** Failure results of test pieces 1 to 9.

### 3.2. The Ultimate Bearing Capacity of the Specimens

In Table 5, the eccentric compression test parameter levels of nine specimens and the test value $N_{ue}$ of ultimate bearing capacity are given. It can be seen from the table that when the slenderness ratio of the specimen is the same, the ultimate bearing capacity shows a decreasing trend with the increase in eccentricity. When the eccentric distance of the specimen is the same, and when the eccentric direction is on the asymmetric axis (x-axis) of the section, the ultimate bearing capacity of the specimen is relatively high.

**Table 5.** Results of concrete material properties test.

| Specimen | $t$/mm | $L$/mm | $e$/mm | Eccentric Direction | $\lambda$ | $N_{ue}$/kN |
|---|---|---|---|---|---|---|
| T-1 | 4 | 600 | 20 | x+ | 7 | 3058.00 |
| T-2 | 4 | 600 | 40 | y+ | 10 | 2290.90 |
| T-3 | 4 | 600 | 60 | Y− | 10 | 1898.70 |
| T-4 | 4 | 1500 | 20 | y+ | 25 | 2409.70 |
| T-5 | 4 | 1500 | 40 | y− | 25 | 1859.50 |
| T-6 | 4 | 1500 | 60 | x+ | 17 | 2036.30 |
| T-7 | 4 | 1800 | 20 | y− | 29 | 2459.50 |
| T-8 | 4 | 1800 | 40 | x+ | 20 | 2340.00 |
| T-9 | 4 | 1800 | 60 | y+ | 29 | 1534.30 |

Notes: $t$ denotes wall thickness of steel tube; $L$ denotes the length of test piece; $e$ denotes eccentricity; $\lambda$ denotes slenderness ratio; $N_{ue}$ denotes test value of the ultimate load.

### 3.3. Load–Strain Curve

Figure 23 shows the correlation curve between the steel strain and the load N at the edge of the compression zone and the tension zone for 1/2 of the column height. It can be seen from the curve that: (1) In the early stage of loading, the longitudinal direction of the edge fibers during the compression and tension process of the strain increase linearly with increasing load. (2) From the beginning of loading to before the limit load, the entire section of the specimen is compressed. When the steel in the compression zone reaches the yield strain, the surface of the steel pipe wall appears with bulging deformation. After that, tensile strain occurs in the tension zone, and the specimen gradually reaches the limit. (3) When the specimen reaches the ultimate bearing capacity, the stress of the steel pipe in the tensile and compressive zone of the specimen reaches its peak at the same time, which proves that the cooperative working performance of the specimen is good. With the increase in strain, the bearing capacity of the specimen shows a gentle decline, and the specimen shows good ductility. (4) During the loading process of the test, the strain in the compression position is larger than that in the tensile position, and the specimen shows a relatively good ductility. Destruction begins when the pressurized area begins to drop out of work.

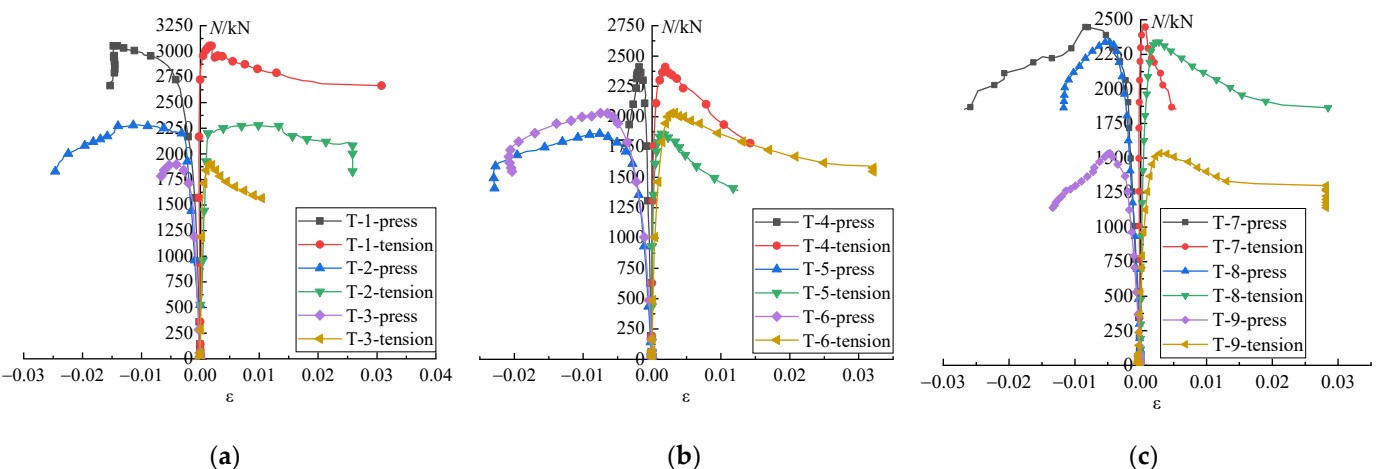

**Figure 23.** Load–strain curves of specimens 1 to 9: (**a**) load–strain curves of specimens 1 to 3; (**b**) load–strain curves of specimens 4 to 6; (**c**) load–strain curves of specimens 7 to 9.

### 3.4. Load–Deflection Curve

Figure 24 shows the N–w relationship curve of specimens No. 1 to No. 9. It can be seen from Figure 24 that at the initial stage of loading of the specimen, the deflection and load show a linear correlation. As the load continues to increase, the horizontal deflection of the column mid-section increases linearly. When approaching the ultimate load, the deflection increases rapidly, and the specimen exhibits obvious bending deformation. When the specimen reaches the ultimate load, the horizontal deflection of the middle of the short column is smaller than that of the middle of the long column.

### 3.5. Strain Distribution of Section in the Column

Figure 25 shows the distribution curve of the section strain along the height of the T-2, T-5, and T-7 columns at different loading stages. It can be seen from the relationship diagram that: (1) Before the specimen reaches the ultimate load, when the specimen is bent and deformed, the distribution of the strain along the height of the middle section of the column basically conforms to the assumption of the plane section, and it can be considered that the deformation of the plane section is maintained. (2) After the specimen reaches the ultimate bearing capacity, the tension area of the middle section of the column still maintains the plane section deformation, but the section deformation of the compression

area is no longer consistent with the assumption of the plane section. (3) With the increase in eccentricity, the neutral axis begins to gradually shift to the direction of the centroid due to the influence of the second-order effect. (4) After the specimen reaches the ultimate bearing capacity, due to the damage of the compression area, the compression position begins to withdraw from work, and the position of the neutralization axis gradually moves towards the centroid of the section.

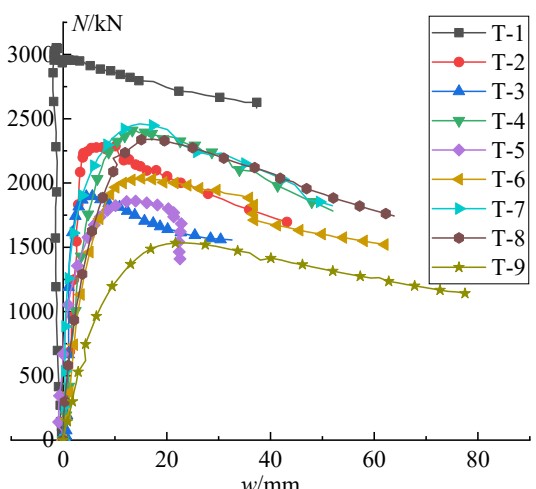

**Figure 24.** Load–deflection curves of specimens 1 to 9.

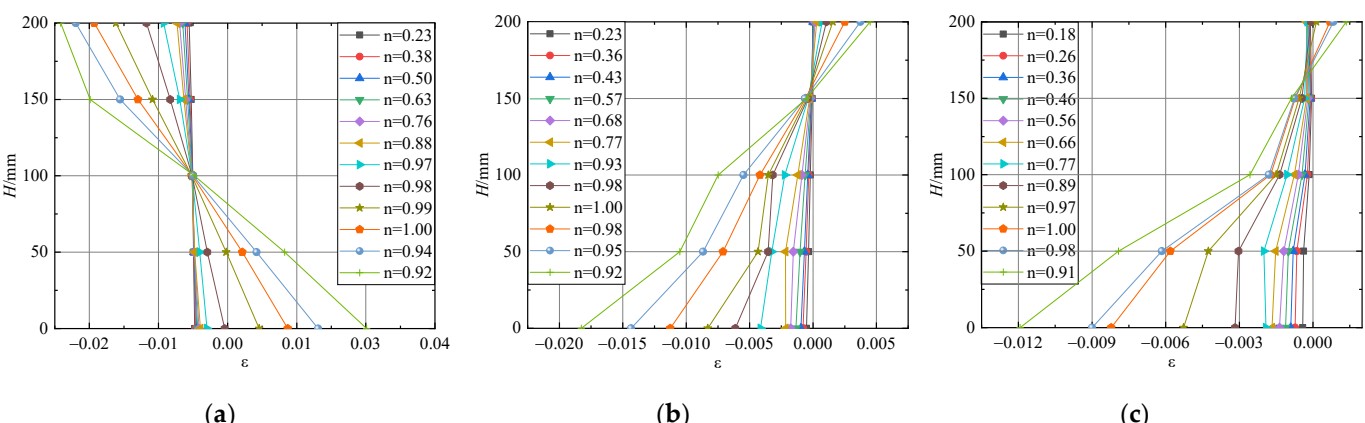

(**a**)                                   (**b**)                                   (**c**)

**Figure 25.** Strain distribution of the middle section of the pieces: (**a**) strain distribution in the column section of No. 2 specimen (e = 40 mm); (**b**) strain distribution in the column section of No. 5 specimen (e = 40 mm); (**c**) strain distribution in the column section of No. 7 specimen (e = 40 mm).

*3.6. Analysis of Test Parameters*

Tables 6–8 are the results of the orthogonal analysis of the test parameters. Through the orthogonal analysis, we can see that: (1) The intuitive analysis of the test results shows the influence of the three test parameters on the eccentric compressive bearing capacity of the T-shaped square concrete-filled steel tube special-shaped column: B (eccentric distance) > C (eccentric direction) > A (length). (2) The range analysis results show that the eccentric distance is the most influential factor on the pressure performance of the eccentricity, followed by the eccentric direction and the length of the specimen The degree is relatively small. (3) The results of variance analysis can show that the fitting results of the general linear models are good, and the results of the variance analysis are also reliable. Among the three test parameters, B (eccentric distance) and C (eccentric direction) have a greater influence on the test value.

**Table 6.** Analysis of test results.

| Specimen | *L*/mm | *e*/mm | Eccentric Direction | $N_{ue}$/kN |
|----------|--------|--------|---------------------|-------------|
| T-1 | 600 | 20 | x+ | 3058.00 |
| T-2 | 600 | 40 | y+ | 2290.90 |
| T-3 | 600 | 60 | y− | 1898.70 |
| T-4 | 1500 | 20 | y+ | 2409.70 |
| T-5 | 1500 | 40 | y− | 1859.50 |
| T-6 | 1500 | 60 | x+ | 2036.30 |
| T-7 | 1800 | 20 | Y− | 2459.50 |
| T-8 | 1800 | 40 | x+ | 2340.00 |
| T-9 | 1800 | 60 | y+ | 1534.30 |
| *K*1 | 2415.867 | 2642.4 | 2478.1 | — |
| *K*2 | 2101.833 | 2163.467 | 2078.3 | — |
| *K*3 | 2111.267 | 1823.1 | 2072.567 | — |
| Range | 314.0333 | 819.3 | 405.5333 | — |
| Rank | 3 | 1 | 2 | 4 |

Notes: *L* denotes the length of the test piece; *e* denotes eccentricity; $N_{ue}$ denotes the test value of ultimate load; *K*1, *K*2, and *K*3 are the average values of the test results of each parameter at the three-parameter levels.

**Table 7.** Mean response analysis of test results.

| Parameter Level | *L*/mm | *e*/mm | Eccentric Direction | Empty Column |
|-----------------|--------|--------|---------------------|--------------|
| 1 | 2416 | 2642 | 2478 | 2151 |
| 2 | 2102 | 2163 | 2078 | 2262 |
| 3 | 2111 | 1823 | 2073 | 2216 |
| Delta | 314 | 819 | 406 | 112 |
| Rank | 3 | 1 | 2 | 4 |

Notes: *L* denotes the length of the test piece; *e* denotes eccentricity; $N_{ue}$ denotes the test value of ultimate load; 1, 2, and 3 represent different parameter values of each factor at three parameter levels, respectively.

**Table 8.** Analysis of variance of test results.

| Parameter | Degrees of Freedom | *Adj SS* | *Adj MS* | Value of *F* | Value of *P* |
|-----------|-------------------|----------|----------|--------------|--------------|
| Length | 2 | 191,487 | 95,744 | 10.14 | 0.090 |
| Eccentricity | 2 | 101,6479 | 508,240 | 53.83 | 0.018 |
| Eccentric direction | 2 | 324,330 | 162,165 | 17.18 | 0.055 |
| Error | 2 | 18,882 | 9441 | — | — |
| Total | 8 | 1,551,178 | — | — | — |
| *S* = 97.1644 | *R-sq* = 98.78% | *R-sq(adjusted)* = 95.13% | 406 | 112 | — |

Notes: *Adj SS* denotes corrected sum of squares; *Adj MS* denotes corrected mean square; *F* denotes the ratio of the between-level variance to the within-level variance of the parameter; *S* denotes the difference between data value and fitting value; *R-sq* denotes goodness of fit, which is the ratio of the regression sum of squares to the total deviation sum of squares; *R-sq(adjusted)* denotes modified goodness of fit.

Figure 26 shows the changing trend of the influence of various factors on the eccentric compressive bearing capacity of the T-shaped CFST composite special-shaped column. With the increase in the eccentric distance, the bearing capacity of the eccentric compression specimen decreases significantly, and with the increase in the length of the specimen and the change in the eccentric direction, the change degree of the eccentric compression specimen is similar. The influence is the largest, followed by the eccentric direction, and finally the length of the specimen.

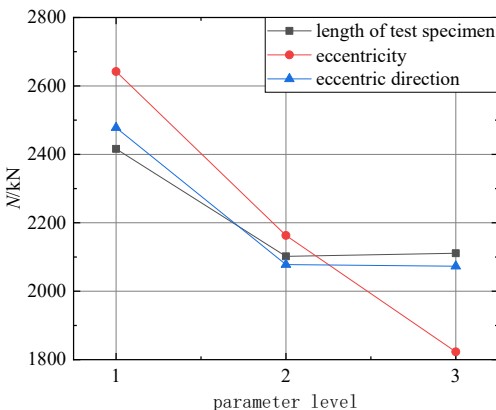

**Figure 26.** Graph of the influence of test parameters.

*3.7. Calculation of Bearing Capacity*

At present, there is no unified design and calculation method for CFST special-shaped columns in the domestic and foreign codes and regulations about CFST. This paper refers to the AISC-LRFD (1999) [33] code of the United States, BS 5400 (1979) [34], the code formulated by the British Standards Association, Eurocode 4 (1994) [35], the code formulated by the European Standards Association, AIJ (1997) [36], the code of the Architectural Society of Japan, the specification for the design and construction of concrete-filled steel tubular structures (CECS 28:90) [37] of China, and the technical specification for concrete-filled steel tube structures (DBJ/t13-51-2010) [38] of Fujian Province of China. Using the calculation formula on the bearing capacity of concrete-filled steel tubular composite columns, the ratio of the calculated value $N_{\mathrm{u}}$ to the test value $N_{\mathrm{ue}}$ is shown in Table 9. Through the comparison of the calculation results of various codes and regulations, it can be found that the calculated values of DBJ/t13-51-2010 and AIJ are the most consistent with the test values, but the dispersion of AIJ is higher than that of DBJ/t13-51-2010; the calculated value of Eurocode 4 is in good agreement with the test value, but slightly worse than DBJ/t13-51-2010 and AIJ, and also BS 5400. The calculated value of CECS is generally consistent with the test value, and the calculated result is unsafe; the calculated value of the AISC specification is in the lowest agreement with the test value, and the calculated result is too safe. It can be seen that using the DBJ/T13-51-2010 specification of the unified theory proposed by Zhong [39], the calculation of the eccentric compressive bearing capacity of the T-shaped square concrete-filled steel tubular composite special-shaped column is in the best agreement with the test results.

**Table 9.** The comparison between the calculated value and the test value of the test piece's eccentric bearing capacity.

| Specimen | $t$/mm | $L$/mm | $e$/mm | $\lambda$ | AISC $\eta$ | DBJ $\eta$ | BS5400 $\eta$ | EC4 $\eta$ | AIJ $\eta$ | CECS $\eta$ |
|---|---|---|---|---|---|---|---|---|---|---|
| T-1 | 4 | 600 | 20 | 7 | 0.718 | 1.018 | 0.821 | 0.904 | 0.948 | 1.149 |
| T-2 | 4 | 600 | 40 | 10 | 0.764 | 1.043 | 0.916 | 1.050 | 1.006 | 1.208 |
| T-3 | 4 | 600 | 60 | 10 | 0.760 | 1.035 | 0.956 | 0.844 | 1.011 | 1.202 |
| T-4 | 4 | 1500 | 20 | 25 | 0.922 | 1.033 | 0.976 | 1.096 | 1.059 | 1.190 |
| T-5 | 4 | 1500 | 40 | 25 | 0.941 | 0.974 | 1.042 | 1.165 | 1.109 | 1.270 |
| T-6 | 4 | 1500 | 60 | 17 | 0.693 | 0.994 | 0.838 | 0.786 | 0.861 | 1.014 |
| T-7 | 4 | 1800 | 20 | 29 | 0.903 | 0.999 | 0.935 | 1.057 | 0.985 | 1.105 |
| T-8 | 4 | 1800 | 40 | 20 | 0.734 | 1.035 | 0.846 | 1.040 | 0.880 | 0.957 |
| T-9 | 4 | 1800 | 60 | 29 | 0.941 | 0.921 | 1.043 | 0.946 | 1.084 | 1.269 |
| $\mu$ | — | — | — | — | 0.820 | 1.006 | 0.930 | 0.987 | 0.994 | 1.151 |
| $\sigma$ | — | — | — | — | 0.098 | 0.037 | 0.079 | 0.118 | 0.081 | 0.102 |

Notes: $t$ denotes wall thickness of the steel tube; $L$ denotes the length of test piece; $e$ denotes eccentricity; $\lambda$ denotes slenderness ratio; $\mu$ denotes mean; $\sigma$ denotes standard deviation; $N_{\mathrm{u}}$ denotes calculated value of ultimate bearing capacity; $N_{\mathrm{ue}}$ denotes test value of ultimate bearing capacity; $\eta$ denotes $N_{\mathrm{u}}/N_{\mathrm{ue}}$.

## 4. Finite Element Modeling and Validation

### 4.1. Finite Element Modeling

According to the Mises yield criterion, obey the isotropic strengthening criterion, follow the corresponding flow law, and select the plastic model provided by ABAQUS for modeling and calculation. The square steel tube adopts the shell element (S4R), and the core concrete part adopts the non-coordinated eight-node linear hexahedron solid element (C3D8I). The contact friction coefficient of the steel is 0.3, and the friction coefficient between the steel and concrete is 0.6. Welds are simulated using binding constraints. Figure 27 shows each component unit and grid division.

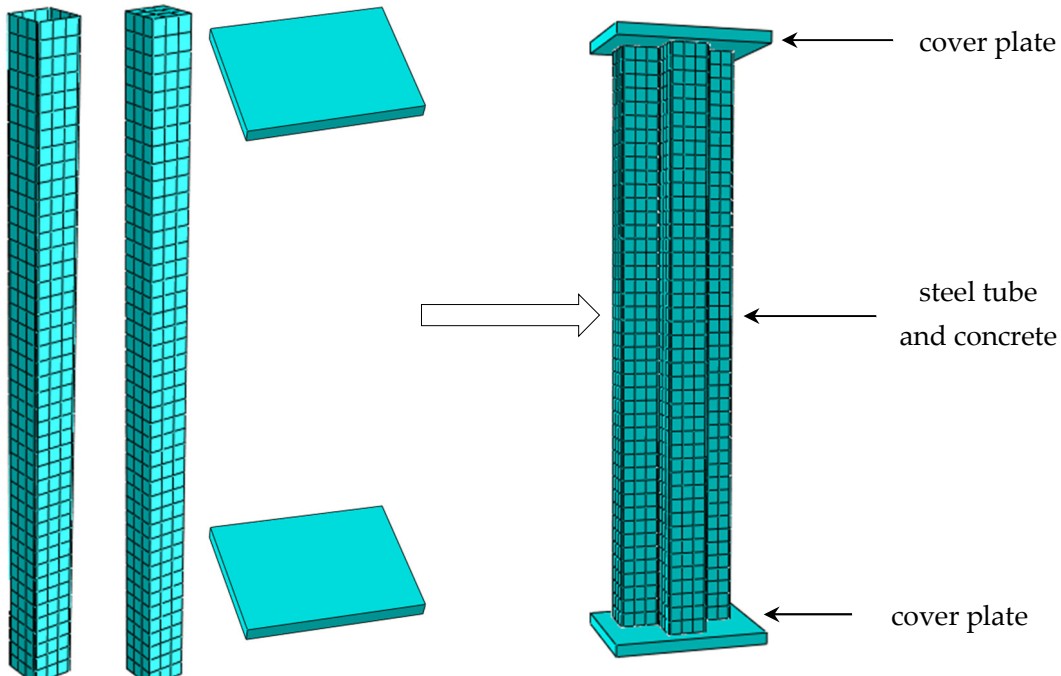

**Figure 27.** Elements and meshing of components.

Using the steel constitutive model proposed by Ding et al. [40], the simplified model of the constitutive relation is shown in Figure 28. Mathematical expressions such as Formula (1) are used.

$$
\sigma_i = \begin{cases}
E_s \varepsilon_i & (\varepsilon_i \leq \varepsilon_y) \\
f_s & (\varepsilon_y < \varepsilon_i \leq \varepsilon_{st}) \\
f_s + \zeta E_s (\varepsilon_i - \varepsilon_{st}) & (\varepsilon_{st} < \varepsilon_i \leq \varepsilon_u) \\
f_u & (\varepsilon_u < \varepsilon_i)
\end{cases}
\tag{1}
$$

where $\zeta = 1/216$; $i$ = the equivalent stress of the steel; $E_s$ = the elastic modulus of the steel, taking $E_s = 2.06 \times 10^5$ MPa; $\varepsilon_i$ = the equivalent strain of the steel; $\varepsilon_y$ = the strain of the steel when it yields; $f_s$ = the yield strength of steel; $\varepsilon_{st}$ = the strain when the steel is strengthened; $\varepsilon_{st} = 12\,\varepsilon_y$; $\varepsilon_u$ = the strain when the steel reaches the ultimate strength; $\varepsilon_u = 120\,\varepsilon_y$; and $f_u$ = the ultimate strength of the steel, taking $f_u = 1.5 f_s$.

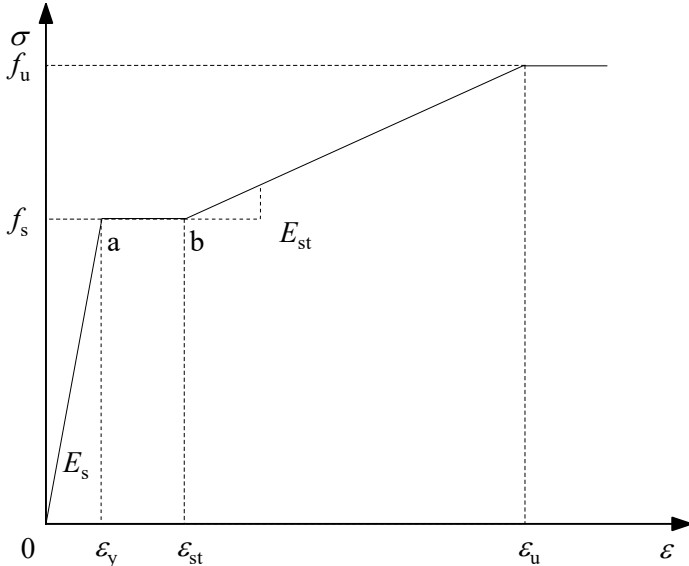

**Figure 28.** Constitutive relation model for steel.

The plastic damage model is used for concrete. The constitutive relation of constrained concrete under compression adopts the modified stress–strain relation of CFST with different strength grades proposed in the literature [40] under uniaxial compression, and the mathematical expression is shown in Formula (2).

$$y = \begin{cases} \frac{A_1 x + (B_1 - 1)x^2}{1 + (A_1 - 2)x + B_1 x^2} & (x \leq 1) \\ \frac{x}{\alpha_1 (x-1)^2 + x} & (x > 1) \end{cases} \tag{2}$$

where $x = \varepsilon/\varepsilon_c$; $y = \sigma/f_c$; $\varepsilon_c$ = the peak compressive strain of concrete; $\varepsilon_c = 383 f_{cu}^{7/18} \times 10^{-6}$; $A_1$ = the ratio of concrete elastic modulus to peak secant modulus; $A_1 = 9.1 f_{cu}^{-4/9}$; and $B_1$ = a physical quantity related to the attenuation of the elastic modulus of the ascending curve. Before the ascending segment $\sigma = 0.4 f_c$, the curve approximates a straight line. At this time, $B_1 = 1.6(A_1 - 1)^2$; $\alpha_1$ is a parameter related to the descending section of the stress–strain curve of concrete under uniaxial compression. For the core concrete under the constraint of steel pipe, $\alpha_1 = f_c^{0.1}/(1.2\sqrt{1+\xi})$, $\xi = A_s f_y/(A_c f_c)$. In the nonlinear finite element analysis of CFST using ABAQUS, good calculation results can be obtained by taking $\alpha_1 = 0.15$; $f_{cu}$ = the compressive strength of the concrete cube; $A_s$ and $f_y$ are the cross-sectional area of the steel pipe and the yield strength of the steel; and $A_c$ and $f_c$ are the cross-sectional area of the concrete and the compressive strength of the concrete axis, where $f_c = 0.4 f_{cu}^{7/6}$.

Constrained concrete plastic damage coefficient $D$ is calculated according to Formula (3), $D_0$ is the damage value of concrete at peak stress; $D_0 = 2.1 - 0.4\ln(f_{cu} + 41)$; for uniaxial compression, the strain $\varepsilon_p$ is the peak compression value strain $\varepsilon_c$; $\varepsilon_c = 383 f_{cu}^{7/18} \times 10^{-6}$; $c_1$, $c_2$ and $c_3$ are calculation parameters; $c_1 = 0.56 - 0.004 f_{cu}$; $c_2 = 1.17 + 4.34 \times 10^{-5} f_{cu}^{2.8}$; and $c_3 = 0.32 + 0.3\ln(f_{cu} - 10)$. Figure 29 shows the constitutive relation curve of concrete.

$$D = \begin{cases} \left[1 - (1 - \varepsilon/\varepsilon_p)^{c_1}\right] D_0 & (\varepsilon \leq \varepsilon_p) \\ 1 - \frac{1 - D_0}{c_2(1 - D_0)(\varepsilon/\varepsilon_p - 1)^{c_3} + 1} & (\varepsilon > \varepsilon_p) \end{cases} \tag{3}$$

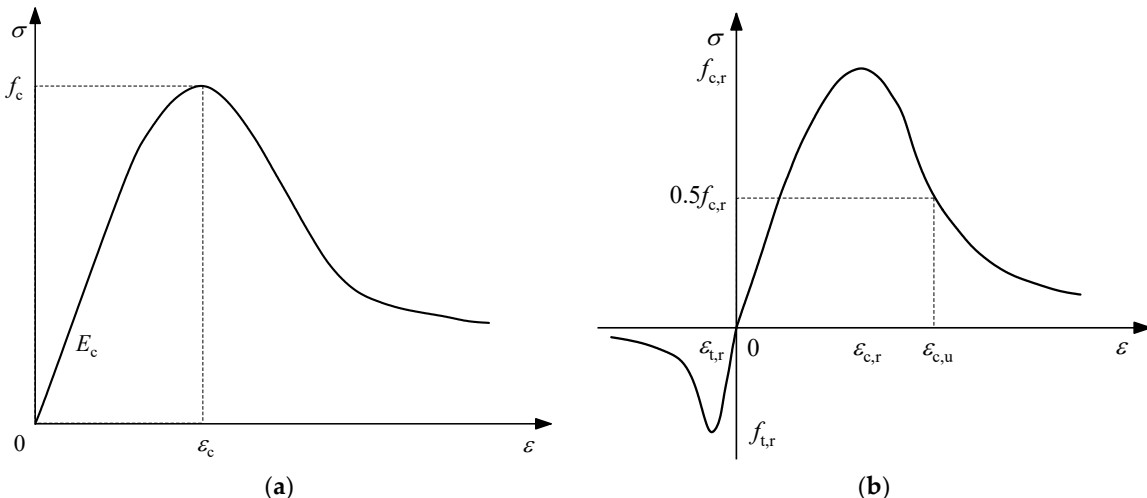

**Figure 29.** Stress–strain curve of concrete: (**a**) uniaxial compressive stress–strain curves for confined concrete; (**b**) uniaxial tensile and compressive stress–strain curves of unconstrained concrete.

The stress–strain relationship of tensile concrete adopts the constitutive curve provided in appendix C of the code for design of concrete structures (GB 50010-2010) [41]. The mathematical expression is as shown in Formula (4); $d_t$ is the uniaxial tensile damage evolution parameter of concrete; $E_c$ is the elastic modulus of concrete, calculated according to $E_c = 9500 f_{cu}^{1/3}$; $\alpha_t$ is the falling section parameter of the concrete uniaxial tensile stress–strain curve, which is taken according to C.2.3 in GB 50010-2010; $\varepsilon_{t,r}$ is the peak tensile strain of concrete corresponding to the representative value of uniaxial tensile strength, which is taken according to C.2.3 in GB 50010-2010; $f_{t,r}$ is the representative value of concrete uniaxial tensile strength, which is taken according to the test results.

$$\sigma = (1 - d_t)E_c\varepsilon \tag{4}$$

$$d_t = \begin{cases} 1 - \rho_t\left[1.2 - 0.2x^5\right] & (x \leq 1) \\ 1 - \dfrac{\rho_t}{\alpha_t(x-1)^{1.7}+x} & (x > 1) \end{cases} \tag{5}$$

$$x = \frac{\varepsilon}{\varepsilon_{t,r}} \tag{6}$$

$$\rho_t = \frac{f_{t,r}}{E_c\varepsilon_{t,r}} \tag{7}$$

*4.2. Reliability Verification of FEM*

Figures 30–38 show the overall stress distribution of No. 1 to No. 9 specimens and the comparison between the failure phenomenon of the finite element model and the experimental phenomenon. From left to right are the failure phenomenon of the test, the failure phenomenon of the finite element model, the failure phenomenon of the steel pipe in the finite element model, and the failure phenomenon of the core concrete in the finite element model.

The failure process and overall stress distribution of the T-1 to T-3 specimen models are shown in Figures 30–32. During the whole loading process, the compression zone begins to yield first. As the load increases, the yield surface gradually expands, and the plastic zone expands. Finally, the surface of the steel tube on the compression side is severely deformed by bulging, followed by bending. The finite element analysis results are in good agreement with the test results.

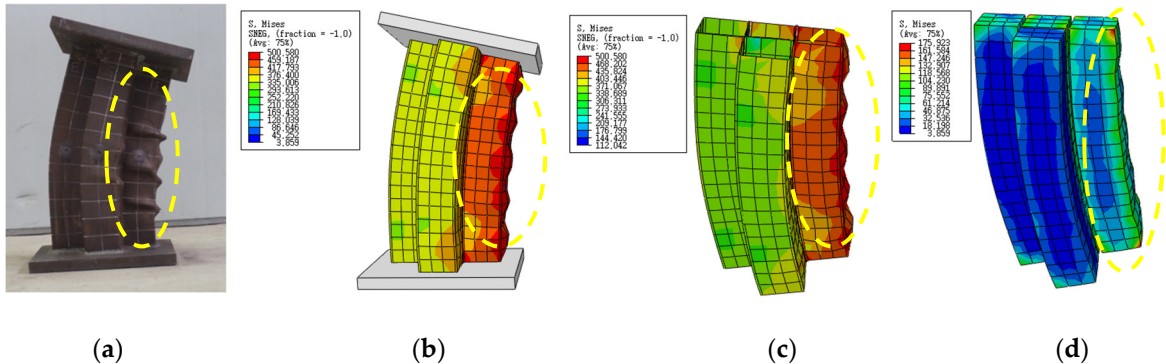

(**a**)  (**b**)  (**c**)  (**d**)

**Figure 30.** Stress distribution and failure phenomenon of finite element model of T-1 specimen: (**a**) Test failure mode; (**b**) Numerical calculation of failure mode; (**c**) Failure modes of steel tube; (**d**) Failure modes of concrete.

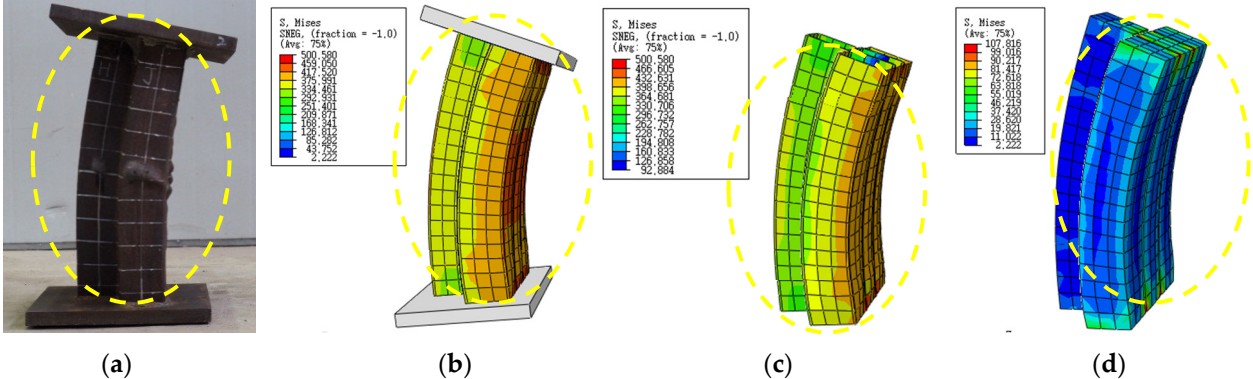

(**a**)  (**b**)  (**c**)  (**d**)

**Figure 31.** Stress distribution and failure phenomenon of finite element model of T-2 specimen: (**a**) Test failure mode; (**b**) Numerical calculation of failure mode; (**c**) Failure modes of steel tube; (**d**) Failure modes of concrete.

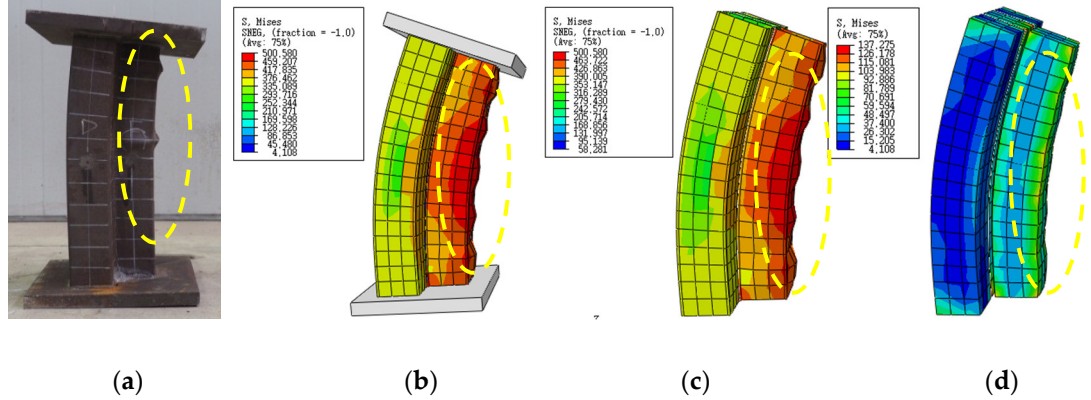

(**a**)  (**b**)  (**c**)  (**d**)

**Figure 32.** Stress distribution and failure phenomenon of finite element model of T-3 specimen: (**a**) Test failure mode; (**b**) Numerical calculation of failure mode; (**c**) Failure modes of steel tube; (**d**) Failure modes of concrete.

The failure process and overall stress distribution of the specimens T-4 to T-9 are shown in Figures 33–38. During the whole loading process, the stress in the compression zone increases the fastest; it first enters the plastic development stage, and the surface of the compression side has bulging deformation. Finally, it was damaged due to bending instability, and the finite element analysis results were in good agreement with the experimental results.

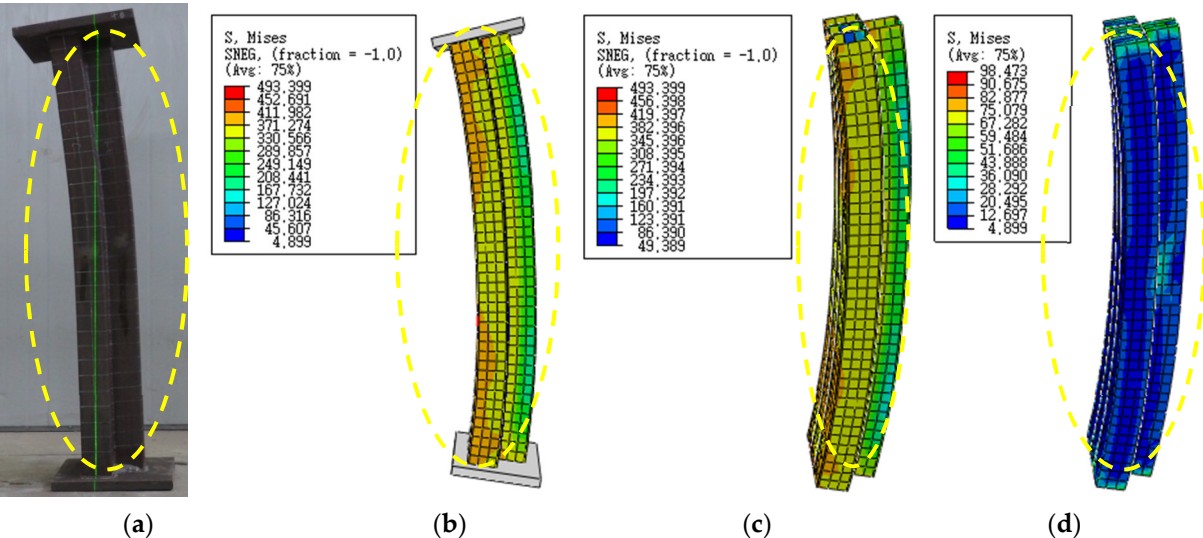

(**a**)         (**b**)         (**c**)         (**d**)

**Figure 33.** Stress distribution and failure phenomenon of finite element model of T-4 specimen: (**a**) Test failure mode; (**b**) Numerical calculation of failure mode; (**c**) Failure modes of steel tube; (**d**) Failure modes of concrete.

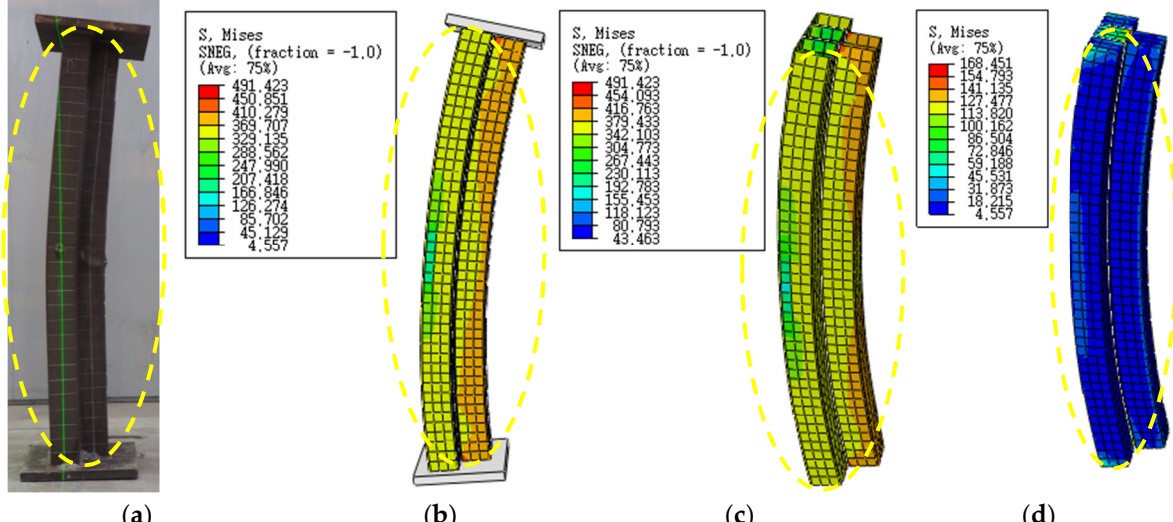

(**a**)         (**b**)         (**c**)         (**d**)

**Figure 34.** Stress distribution and failure phenomenon of finite element model of T-5 specimen: (**a**) Test failure mode; (**b**) Numerical calculation of failure mode; (**c**) Failure modes of steel tube; (**d**) Failure modes of concrete.

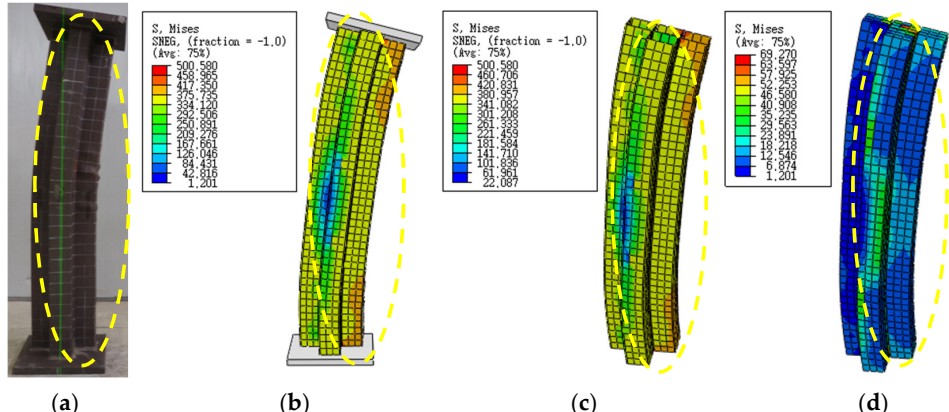

**Figure 35.** Stress distribution and failure phenomenon of finite element model of T-6 specimen: (**a**) Test failure mode; (**b**) Numerical calculation of failure mode; (**c**) Failure modes of steel tube; (**d**) Failure modes of concrete.

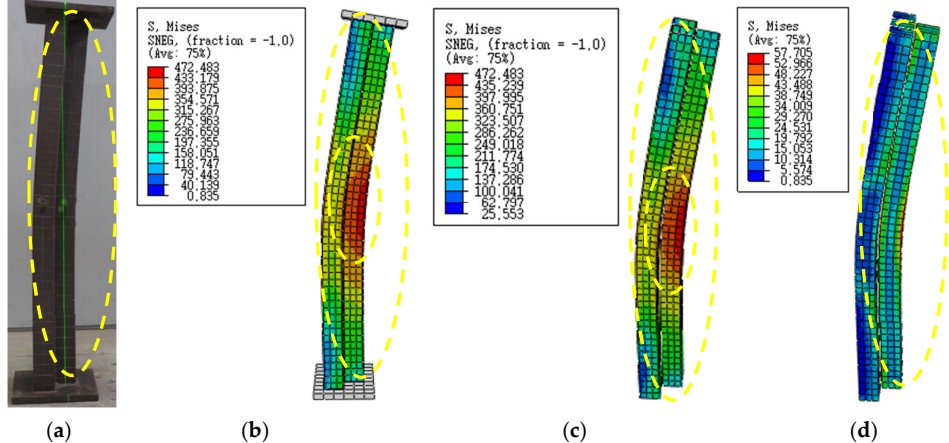

**Figure 36.** Stress distribution and failure phenomenon of finite element model of T-7 specimen: (**a**) Test failure mode; (**b**) Numerical calculation of failure mode; (**c**) Failure modes of steel tube; (**d**) Failure modes of concrete.

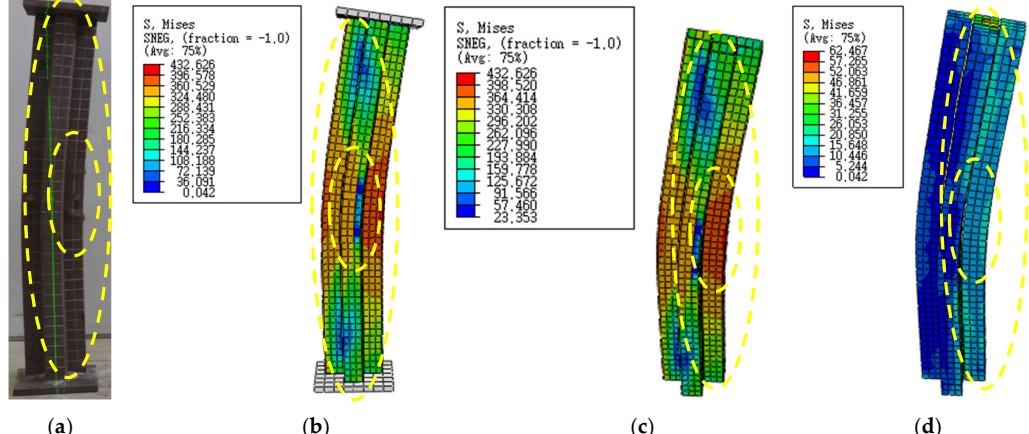

**Figure 37.** Stress distribution and failure phenomenon of finite element model of T-8 specimen: (**a**) Test failure mode; (**b**) Numerical calculation of failure mode; (**c**) Failure modes of steel tube; (**d**) Failure modes of concrete.

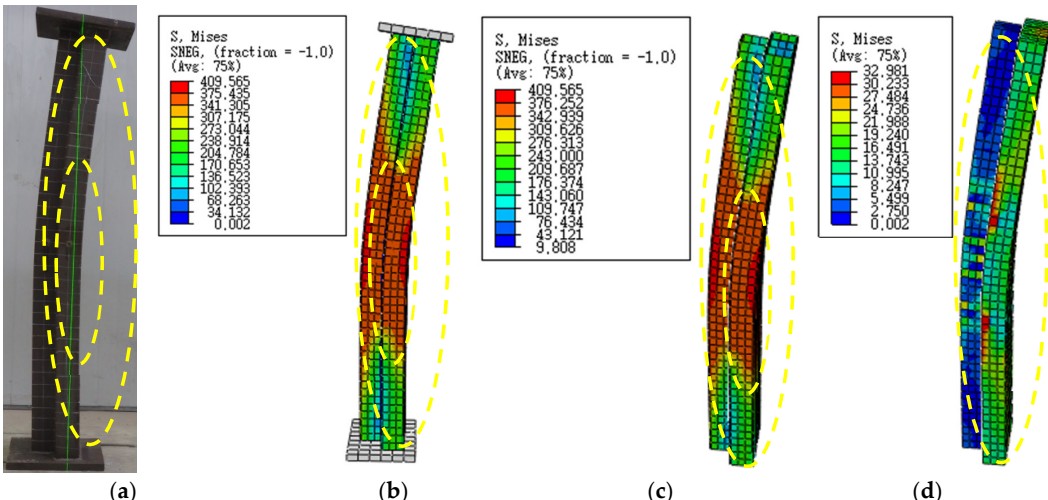

**Figure 38.** Stress distribution and failure phenomenon of finite element model of T-9 specimen: (**a**) Test failure mode; (**b**) Numerical calculation of failure mode; (**c**) Failure modes of steel tube; (**d**) Failure modes of concrete.

The ultimate bearing capacity values and corresponding parameters of specimens 1–9 obtained through finite element analysis calculations are listed in Table 10. The ratio between the calculated value of finite element simulation and the test value is calculated in the table.

**Table 10.** Ultimate bearing capacity of T-shaped concrete-filled square steel tubular composite special-shaped column under eccentric compression.

| Specimen | $L$/mm | $e$/mm | Eccentric Direction | $\lambda$ | $N_{uf}$/kN | $N_{ue}$/kN | $N_{uf}/N_{ue}$ |
|---|---|---|---|---|---|---|---|
| T-1 | 600 | 20 | x+ | 7 | 3042.57 | 3058.00 | 0.99 |
| T-2 | 600 | 40 | y+ | 10 | 2348.86 | 2290.90 | 1.03 |
| T-3 | 600 | 60 | y− | 10 | 1818.53 | 1898.70 | 0.96 |
| T-4 | 1500 | 20 | y+ | 25 | 2583.20 | 2409.70 | 1.07 |
| T-5 | 1500 | 40 | y− | 25 | 1943.16 | 1859.50 | 1.04 |
| T-6 | 1500 | 60 | x+ | 17 | 1925.75 | 2036.30 | 0.95 |
| T-7 | 1800 | 20 | y− | 29 | 2358.61 | 2459.50 | 0.96 |
| T-8 | 1800 | 40 | x+ | 20 | 2205.27 | 2340.00 | 0.94 |
| T-9 | 1800 | 60 | y+ | 29 | 1574.10 | 1534.30 | 1.03 |

Notes: $L$ denotes the length of the test piece; $e$ denotes eccentricity; $\lambda$ denotes slenderness ratio; $N_{uf}$ denotes the simulated value of bearing capacity; $N_{ue}$ denotes the test value of the ultimate load. The steel strength grade of the test piece is Q235B, the concrete strength grade is C30, and the wall thickness of the steel pipe is 4 mm.

### 4.3. Load–Strain Curve of FEM

Figure 39 shows the comparison between the load–strain of the nine specimen models and the test results, Ei represents the test result of the i-th specimen, and FEi represents the finite element calculation result of the i-th specimen. Through the test and the finite element load–strain curve, it can be found that in the elastic stage, the longitudinal strain of the steel pipe increases gradually with the increase in load. After entering the plastic stage, the strain growth accelerates, the steel deforms greatly, and the bearing capacity of the component decreases slowly. The experimental results are in good agreement with the finite element calculation results.

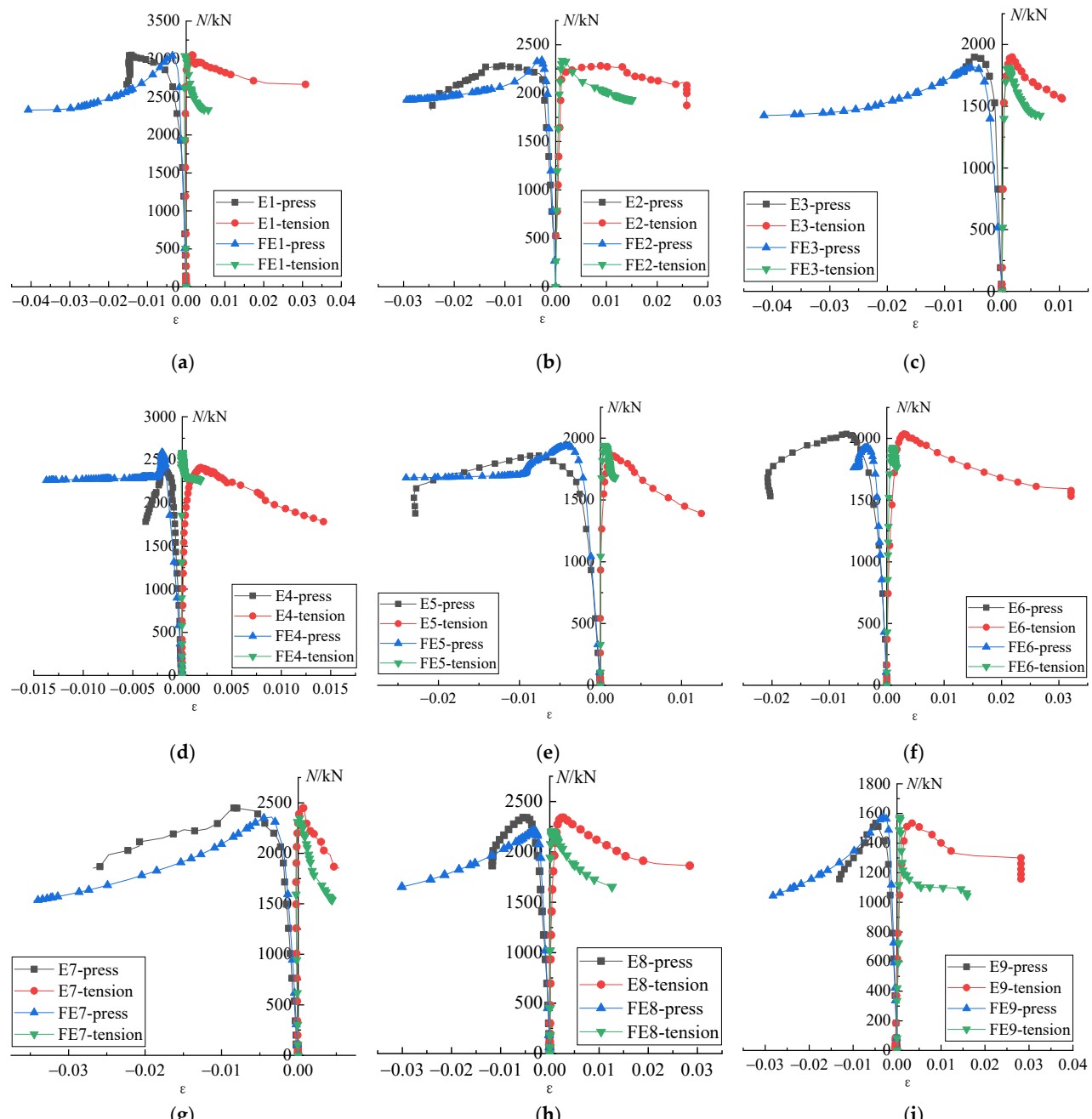

**Figure 39.** Load–strain curve of FEM: (**a**) load–strain curve of T-1; (**b**) load–strain curve of T-2; (**c**) load–strain curve of T-3; (**d**) load–strain curve of T-4; (**e**) load–strain curve of T-5; (**f**) load–strain curve of T-6; (**g**) load–strain curve of T-7; (**h**) load–strain curve of T-8; (**i**) load–strain curve of T-9.

*4.4. Load–Deflection Curve of FEM*

Figure 40 shows the finite element calculation results and test results of the load N and the horizontal deflection w of the T-1 to T-9 specimens. Ei represents the experimental value of the *i*-th specimen, and FEi represents the finite element calculation value of the *i*-th specimen. Before the specimen reaches the ultimate load, the horizontal deflection of the specimen is small. After reaching the ultimate load, the horizontal deflection of the specimen gradually increased with the decrease in load; after reaching the ultimate load, the specimen showed good ductility; the finite element calculation curve was in good agreement with the experimental curve.

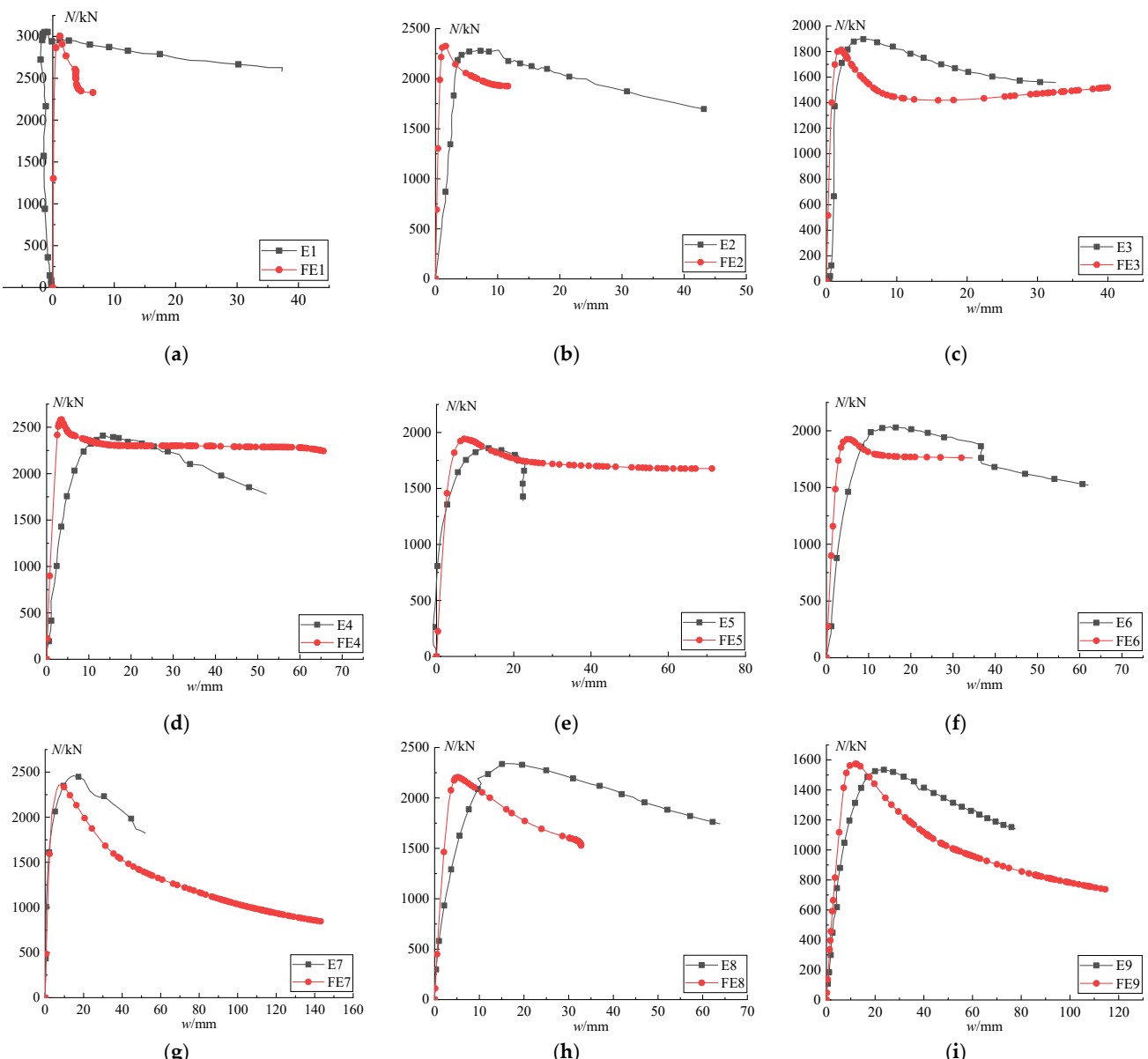

**Figure 40.** Load–deflection curve of FEM: (**a**) load–deflection curve of T-1; (**b**) load–deflection curve of T-2; (**c**) load–deflection curve of T-3; (**d**) load–deflection curve of T-4; (**e**) load–deflection curve of T-5; (**f**) load–deflection curve of T-6; (**g**) load–deflection curve of T-7; (**h**) load–deflection curve of T-8; (**i**) load–deflection curve of T-9.

## 5. Conclusions

The effects of specimen length, eccentric distance, and eccentric direction on the eccentric compression performance of T-shaped concrete-filled square steel tubular composite special-shaped columns are studied through an eccentric compression test, and the calculated values in the code are compared with the experimental values. The numerical analysis model of a T-shaped concrete-filled square steel tubular composite special-shaped column is established by using the general finite element numerical analysis program, ABAQUS. The modeling calculation of nine specimens in the test is carried out to verify the test results and theoretical analysis results.

1.   The failure mode of the T-shaped short column specimen is mainly strength failure, and the long column specimen is mainly bending instability failure. The ultimate bearing capacity of the short column under eccentric compression is higher than

that of the long column. The bending deformation of the specimen is similar to the sinusoidal half wave curve, and there is no torsional deformation. In the process of the eccentric compression failure of the T-shaped concrete-filled square steel tubular composite special-shaped column, the strain in the compression area is large, and the steel pipe wall in the compression area first begins to yield and enters the plastic stage, resulting in large bulging deformation and bending deformation.

2. During the stress process of the specimen, the strain distribution on the section in the middle of the column is consistent with the plane section assumption. The compression area yields before the tension area. The deformation of tensile and compressive stress areas is relatively coordinated, the cooperative working performance of all parts of the specimen is good, and the specimen has good ductility.

3. The eccentricity has the greatest influence on the mechanical properties of the specimen under bias pressure, followed by the eccentricity direction, and finally the specimen length. The influence of eccentricity and eccentricity direction on the mechanical properties of the specimen under bias pressure is more significant than the specimen length.

4. The finite element calculation results are in good agreement with the experimental results. The simulated value of ultimate bearing capacity is basically consistent with the test value, and the failure mode of the finite element specimen model is also consistent with the test. The finite element calculation model of the T-shaped concrete-filled square steel tubular composite special-shaped column has good reliability and can be used as the basis of theoretical calculation and analysis.

5. Comparing the calculation results and test results of six codes at home and abroad, it is found that the calculated values of the concrete-filled steel tubular bearing capacity formula recommended by DBJ / T13-51-2010 and AIJ are in good agreement with the test values, but the calculation results of DBJ / T13-51-2010 are less discrete.

**Author Contributions:** Q.L., methodology, conceptualization, writing—review and editing; Z.L., writing—review and editing, funding acquisition; X.Z., methodology, conceptualization, writing—review and editing, funding acquisition; Z.W., writing—review and editing. All authors have read and agreed to the published version of the manuscript.

**Funding:** This research was supported by the Major Scientific and Technological Innovation Projects of Shandong Province, grant number 2021CXGC011204; the Natural Science Foundation of Shandong Province, grant number ZR2020QE247; the Research and development project of Housing and UrbanRural Development of Shandong Province, grant number 2021-K5-14 and the Research Fund for the Doctoral Program of Shandong Jianzhu University, grant number X19035Z.

**Institutional Review Board Statement:** Not applicable.

**Informed Consent Statement:** Not applicable.

**Data Availability Statement:** The data presented in this study are available on request from the authors.

**Conflicts of Interest:** The authors declare no conflict of interest.

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
