# Peer review of "Experimental Study and Finite Element Calculation of the Behavior of Special T-Shaped Composite Columns with Concrete-Filled Square Steel Tubulars under Eccentric Loads"

_buildings, doi:10.3390/buildings12101756_

Round 1
Reviewer 1 Report
The eccentric compression performance of T-shaped concrete-filled square steel tubular composite special-shaped columns was investigated by means of experiment and numerical simulation in the manuscript. The effects of various parameters including the slenderness ratio, eccentric distance, and the eccentric direction on the compression behavior of the T-shaped concrete-filled square steel tubular composite special-shaped columns were discussed. Some corresponding conclusions are drawn, which have important research significance for guiding engineering practice. However, the following modifications are suggested.
1. To more clearly understand the actual innovative aspects and benefits of the special-shaped column members, a more detailed discussion of the state of the art must be added, including recent journal contributions not considered in the current version of the manuscript.
2. The words "To avoid facility and failure," in line 36 can be deleted. The content of this sentence is not well understood. The sentence " and a formula for calculating the bearing capacity of axial compression is established" in lines 42 to 43 can be changed to " And the calculation formula of axial bearing capacity is established" independently. So, English of the article needs further improvement
3. Is it necessary to set two displacement measuring devices on the upper and lower cover plates of the column in Picture 12? And this measuring device is not seen in the test picture. The schematic diagram of the measuring device shown in Figure 12 needs to be modified to be consistent with the actual test situation.
4. The interface boundary condition is the key factor to determine the simulation result, in section 4.1, “The contact friction coefficient between steel is 0.3, and the friction coefficient between steel and concrete is 0.6”, on what basis are these values determined?
5. It is recommended to further refine the conclusions.
Author Response
Dear Reviewer,
Many thanks for your time in reading our manuscript and for giving us your insightful suggestions. All these are of great importance for improving the quality of this manuscript. We do value these suggestions and appreciate your constructive comments on our manuscript entitled “Experimental study and finite element calculation on the be-havior of special T-shaped composite columns with con-crete-filled square steel tubulars under eccentric loads” (Manuscript ID: buildings-1964021).
We have responded to your comments one by one in the document. At the same time, we have made modifications to the manuscript in response to your suggestions and comments. We hope that all these changes fulfill the requirements to make the revised manuscript acceptable for publication in Buildings.
Thank you very much for your comments and suggestions.
Best regards,
All Authors
Authors’ response to Reviewers’ comments
The eccentric compression performance of T-shaped concrete-filled square steel tubular composite special-shaped columns was investigated by means of experiment and numerical simulation in the manuscript. The effects of various parameters including the slenderness ratio, eccentric distance, and the eccentric direction on the compression behavior of the T-shaped concrete-filled square steel tubular composite special-shaped columns were discussed. Some corresponding conclusions are drawn, which have important research significance for guiding engineering practice. However, the following modifications are suggested.
Point 1. To more clearly understand the actual innovative aspects and benefits of the special-shaped column members, a more detailed discussion of the state of the art must be added, including recent journal contributions not considered in the current version of the manuscript.
Authors’ Reply:
Thanks for your suggestion. We supplement the research work on concrete-filled square steel tubular composite special-shaped columns in a recent journal. This article has great reference value. Wang proposed a new type of special-shaped concrete-filled square steel tube composite column (SS-CFSST), which is composed of multiple square steel tubes connected by steel hoops. The experimental study and finite element analysis of concrete-filled square steel tubular special-shaped composite columns with steel hoops under axial load are introduced.
Wang, Z.; Liu, Z.; Zhou, X. Experimental Investigation of Special-Shaped Concrete-Filled Square Steel Tube Composite Columns with Steel Hoops under Axial Loads. Materials 2022, 15, 4179. https://doi.org/10.3390/ma15124179(References [9])
Point 2. The words "To avoid facility and failure," in line 36 can be deleted. The content of this sentence is not well understood. The sentence " and a formula for calculating the bearing capacity of axial compression is established" in lines 42 to 43 can be changed to " And the calculation formula of axial bearing capacity is established" independently. So, English of the article needs further improvement
Authors’ Reply:
According to your suggestion, the sentence “ To avoid facility and failure,” has been deleted. The sentence “ and a formula for calculating the bearing capacity of axial compression is established” in lines 42 to 43 has been changed to " And the calculation formula of axial bearing capacity is established" independently.
Point 3. Is it necessary to set two displacement measuring devices on the upper and lower cover plates of the column in Picture 12? And this measuring device is not seen in the test picture. The schematic diagram of the measuring device shown in Figure 12 needs to be modified to be consistent with the actual test situation.
Authors’ Reply:
As you have found, the measuring devices at the top and bottom cover plates in Figure 12 are improperly indicated. The schematic diagram of the measuring device shown in Figure 12 has been modified to conform to the actual test situation.
Point 4. The interface boundary condition is the key factor to determine the simulation result, in section 4.1, “The contact friction coefficient between steel is 0.3, and the friction coefficient between steel and concrete is 0.6”, on what basis are these values determined?
Authors’ Reply:
According to the literature and tutorial materials inquired, the static friction coefficient between steels without lubrication can be 0.15. Considering that the cold-formed steel pipe only has simple impurity and rust removal, and the steel surface is rough and slightly rusted, the friction coefficient between the steels is taken as 0.3. Through mutual verification of several pieces of literature, when the friction coefficient between ordinary steel plate and concrete is 0.6, better convergence and calculation results can be obtained in the contact analysis between steel plate and concrete.
Point 5. It is recommended to further refine the conclusions.
Authors’ Reply:
According to your suggestions, the conclusion has been modified to make it more perfect. It is revised as follows:
- The failure mode of the T-shaped short column specimen is mainly section strength failure, and the long column specimen is mainly bending instability failure. The ultimate bearing capacity of the short column under eccentric compression is higher than that of the long column. The bending deformation of the specimen is similar to the sinusoidal half wave curve, and there is no torsional deformation. In the process of eccentric compression failure of T-shaped concrete-filled square steel tubular composite special-shaped column, the strain in the compression area is large, and the steel pipe wall in the compression area first begins to yield and enters the plastic stage, resulting in large bulging deformation and bending deformation.
- During the stress process of the specimen, the strain distribution on the section in the middle of the column is consistent with the plane section assumption. The compression area yields before the tension area. The deformation of tensile and compressive stress areas is relatively coordinated, the cooperative working performance of all parts of the specimen is good, and the specimen has good ductility.
- The eccentricity has the greatest influence on the mechanical properties of the specimen under bias pressure, followed by the eccentricity direction, and finally the specimen length. The influence of eccentricity and eccentricity direction on the mechanical properties of the specimen under bias pressure is more significant than the specimen length.
- The finite element calculation results are in good agreement with the experimental results. The simulated value of ultimate bearing capacity is basically consistent with the test value, and the failure mode of the finite element specimen model is also consistent with the test. The finite element calculation model of T-shaped concrete-filled square steel tubular composite special-shaped column has good reliability and can be used as the basis of theoretical calculation and analysis.
- Comparing the calculation results and test results of six codes at home and abroad, it is found that the calculated values of the concrete-filled steel tubular bearing capacity formula recommended by DBJ / T13-51-2010 and AIJ are in good agreement with the test values, but the calculation results of DBJ / T13-51-2010 are less discrete.

Reviewer 2 Report
Based on the improvement and optimization of the special-shaped section form of concrete-filled steel tube, this manuscript proposes a new type of composite special-shaped concrete-filled square steel tube column, carries out relevant analysis and tests, and uses advanced finite element numerical simulation method to model and calculate, and compares with the test results. However, two issues need further discussion.
First, in the section of special-shaped column shown in Figures 7 and 8, the core concrete in two adjacent square steel tubes is divided by two layers of steel tube walls. The restraining effect of the middle position of the side length of the rectangular section on the core concrete is weaker than that of the corner, and the restraining effect of the two layers of the steel pipe wall is not necessarily much better than that of the one layer of the steel pipe wall. There is excess steel here that does not play its role, but it may make some contributions to the vertical bearing capacity.
Second, only the section of 100 mm × The 100 mm steel pipe column has been tested, and the length of the test pieces ranges from 600 mm to 1800 mm. The number of test pieces in the test is not enough. If conditions permit, the size range of the test pieces can be expanded, such as by increasing the section size of the steel pipe or increasing the length of the column. The test results obtained from more specimens can analyze the eccentric compression performance of special-shaped columns more accurately, and can also provide a more meaningful reference for the design of practical projects.
Author Response
Dear reviewer,
Many thanks for your time in reading our manuscript and for giving us your insightful suggestions. All these are of great importance for improving the quality of this manuscript. We do value these suggestions and appreciate your constructive comments on our manuscript entitled “Experimental study and finite element calculation on the behavior of special T-shaped composite columns with concrete-filled square steel tubulars under eccentric loads” (Manuscript ID: buildings-1964021).
We have explained and responded to your comments one by one in the document. We hope that all these explanations and changes fulfill the requirements to make the revised manuscript acceptable for publication in Buildings.
Thank you very much for your comments and suggestions.
Best regards,
All Authors
Authors’ response to Reviewers’ comments
Based on the improvement and optimization of the special-shaped section form of concrete-filled steel tube, this manuscript proposes a new type of composite special-shaped concrete-filled square steel tube column, carries out relevant analysis and tests, and uses advanced finite element numerical simulation method to model and calculate, and compares with the test results. However, two issues need further discussion.
Point 1. In the section of special-shaped column shown in Figures 7 and 8, the core concrete in two adjacent square steel tubes is divided by two layers of steel tube walls. The restraining effect of the middle position of the side length of the rectangular section on the core concrete is weaker than that of the corner, and the restraining effect of the two layers of the steel pipe wall is not necessarily much better than that of the one layer of the steel pipe wall. There is excess steel here that does not play its role, but it may make some contributions to the vertical bearing capacity.
Authors’ Reply:
As you said, one of the two steel pipe walls between the concrete is redundant, but it contributes to the vertical bearing capacity. However, the main reason for our design is that the finished rectangular square steel pipe is easy to purchase locally and convenient to process. In this way, multiple cutting and welding of steel plates are avoided, and the influence of residual stress on the mechanical properties of steel is reduced.
Point 2. Only the section of 100 mm × The 100 mm steel pipe column has been tested, and the length of the test pieces ranges from 600 mm to 1800 mm. The number of test pieces in the test is not enough. If conditions permit, the size range of the test pieces can be expanded, such as by increasing the section size of the steel pipe or increasing the length of the column. The test results obtained from more specimens can analyze the eccentric compression performance of special-shaped columns more accurately, and can also provide a more meaningful reference for the design of practical projects.
Authors’ Reply:
It can be seen from the test results that the bearing capacity of T-shaped concrete-filled square steel tubular composite special-shaped columns is very good. Unfortunately, due to the limitation of time, experimental funds, and the loading capacity and height of the press equipment in the laboratory, we have not designed more specimens for experimental research. However, we use the finite element calculation model to supplement the analysis. In the future, if we have more time and financial support, we will design more specimens with different parameters for testing.

Reviewer 3 Report
It is an interesting work and the information is clear. Perhaps the biggest criticism comes from the use of normal materials and not high-performance ones: high strength concrete or steel. Of course, reducing the thickness of the steel plates is recommended to fully exploit the advantages of CFT columns.
Author Response
Dear reviewer,
Many thanks for your time in reading our manuscript and for giving us your insightful suggestions. All these are of great importance for improving the quality of this manuscript. We do value these suggestions and appreciate your constructive comments on our manuscript entitled “Experimental study and finite element calculation on the behavior of special T-shaped composite columns with concrete-filled square steel tubulars under eccentric loads” (Manuscript ID: buildings-1964021).
We have explained and responded to your comments one by one in the document. We hope that all these explanations and changes fulfill the requirements to make the revised manuscript acceptable for publication in Buildings.
Thank you very much for your comments and suggestions.
Best regards,
All Authors
Authors’ response to Reviewers’ comments
It is an interesting work and the information is clear. Perhaps the biggest criticism comes from the use of normal materials and not high-performance ones: high strength concrete or steel. Of course, reducing the thickness of the steel plates is recommended to fully exploit the advantages of CFT columns.
Authors’ Reply:
First of all, thank you for your interest in our research work. It can be seen from the test results that the bearing capacity of T-shaped concrete-filled square steel tubular composite special-shaped columns is very good. Unfortunately, due to the limited loading capacity and height of the press equipment in the laboratory, it cannot provide greater loading force and higher space. The specimens made of steel and concrete with higher strength will have higher bearing capacity. However, due to the limitations of test equipment and site, these specimens can not fully exert their ultimate bearing capacity on the test equipment. Later, if conditions permit, we will use higher-strength steel and concrete design specimens for testing. And the finite element simulation is used to supplement the analysis.
